# DEBIASING CLIP: INTERPRETING AND CORRECTING BIAS IN ATTENTION HEADS

## ABSTRACT

Multimodal models like CLIP have gained significant attention due to their remarkable zero-shot performance across various tasks. However, studies have revealed that CLIP can inadvertently learn spurious associations between target variables and confounding factors. To address this, we introduce LOCATE-THEN-CORRECT (LTC), a contrastive framework that identifies spurious attention heads in Vision Transformers via mechanistic insights and mitigates them through targeted ablation. Furthermore, LTC identifies salient, task-relevant attention heads, enabling the integration of discriminative features through orthogonal projection to improve classification performance. We evaluate LTC on benchmarks with inherent background and gender biases, achieving over a $> 50\%$ gain in worst-group accuracy compared to non-training post-hoc baselines. Additionally, we visualize the representation of selected heads and find that the presented interpretation corroborates our contrastive mechanism for identifying both spurious and salient attention heads.

## 1 INTRODUCTION

Rapid advancements in multimodal foundation models like CLIP (Radford et al., 2021; Singha et al., 2024; Fan et al., 2024; Zhang et al., 2025) have enabled remarkable zero-shot learning capabilities across various tasks. However, these models often inherit undesirable biases due to spurious correlations present in their extensive training datasets or imbalanced data distributions (Mao et al., 2023; Alabdulmohsin et al., 2024). Biases include associations between target classes and confounding attributes, e.g., background (Du et al., 2022; Zhang & Ré, 2022; Sagawa et al., 2019; Wang et al., 2024) or gender (Xiao et al., 2024; Hall et al., 2024; Nadeem et al., 2025), which largely degrade performance in underrepresented subgroups and perpetuate harmful stereotypes.

Recently, Gandelsman et al. (2023) proposed to ground visual representations of intermediate attention heads in CLIP's vision Transformer (ViT) (Dosovitskiy, 2020) onto a set of natural language statements. In addition to enhanced interpretability, this enables locating certain components in the model that may encode unwanted biases. However, a downside to the proposed framework is the need for extensive manual effort in summarizing the representation from the set of statements, which often can be inconclusive.

Existing approaches to debiasing vision models often rely on extensive fine-tuning (Zhang & Ré, 2022; Sagawa et al., 2019; Wortsman et al., 2022), which can be computationally prohibitive for large foundation models. Training-free methods, on the other hand, may include orthogonal projection in the image (Adila et al., 2023) or text (Chuang et al., 2023) representation space. In contrast, we propose to perform debiasing only on specific attention heads, while leaving the rest of the model untouched. Our framework, **Locate-Then-Correct (LTC)**, first identifies attention heads that strongly encode spurious attributes and target class features. These activations are then subjected to debiasing procedures, either by removing spurious associations or injecting class-discriminative features through orthogonal projection. LTC employs a "*diagnose-then-correct*" approach to address bias at a granular level within the vision model.

Our work provides a concrete demonstration of how mechanistic insights can be translated into practical tools for debiasing, addressing a pressing challenge in modern machine learning. Beyond improving robustness, our approach offers significantly greater interpretability than existing methods, enabling a clearer understanding of why specific model components should be corrected and how such corrections impact behavior.

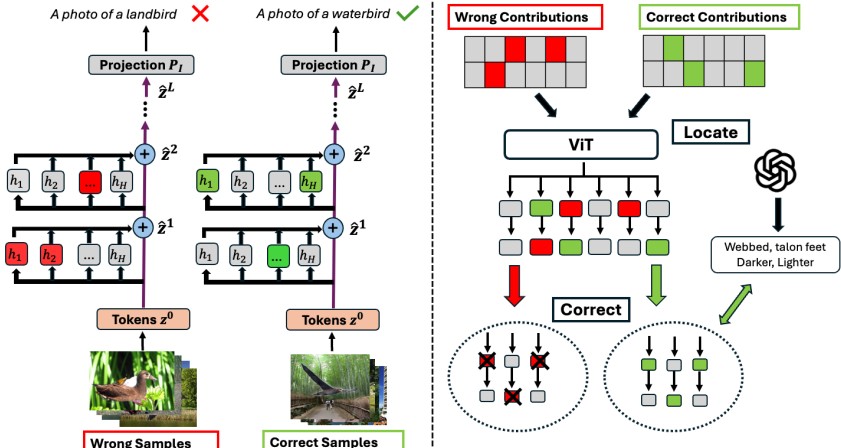

Figure 1: **Left:** Linear decomposition of image representations into individual attention head activations (Elhage et al., 2021). Spurious states (background: land) activate more strongly on images with opposing attributes, whereas target states (class: waterbird) activate on images with matching attributes. **Right:** LTC identifies and corrects these states: mean-ablation mitigates spurious states, while knowledge injection enhances target states.

## 2 RELATED WORK

**Bias in Vision.** Improving robustness against undesirable spurious correlations in vision foundation models is an active researched area. These works include training methods which are split between requiring supervised group labels (Sagawa et al., 2019; Zhang & Ré, 2022) and inferring group labels in an unsupervised manner (Liu et al., 2021; Nam et al., 2020; Sohoni et al., 2020). Non-training methods include utilizing orthogonal projections to erase attribute knowledge and enhance discriminative properties (Adila et al., 2023) or remove spurious relationships attributes (Chuang et al., 2023). Kim et al. (2024) proposes to incorporate detected bias keywords into class prompts with large language models (LLM) to improve robustness. However, there are no explainable insights as to why these methods work. In contrast, our framework only performs corrective measures on specific decomposed components of the model and enables fine-grained interpretation.

**Interpretability in Transformers.** Transformer (Vaswani, 2017) interpretability has been studied at various granularities, including neurons (Olah et al., 2017; Bau et al., 2020; Goh et al., 2021; Shaham et al., 2024) and attention layers or heads (Gandelsman et al., 2023; Yeo et al., 2024; Vig et al., 2020). Elhage et al. (2021) showed that Transformers can be viewed as a linear combination of information from attention heads in the residual stream, shaping the final representation. Leveraging this decomposability, Nostalgebraist (2020); Jiang et al. (2024) examined localized predictions of intermediate states. Our work extends these efforts by identifying and interpreting specific states to enhance robustness and explain their effectiveness.

## 3 BACKGROUND

We start by looking at the architecture of the CLIP (Radford et al., 2021) model and how a classification prediction is made. We primarily focus only on the ViT architecture due to its decomposable nature. CLIP consists of an image encoder $E_I$, and a text encoder $E_T$. Given an image, $I$, the prediction logit $S_y$ for a target class, $y \in Y$ is computed using the cosine similarity between the projected image and text representation $P_I(E_I(I))$ and $P_T(E_T(y))$ respectively. $P_I$ and $P_T$ denotes the image and text projection layer which are parameterized separately.

$$S_y = \langle P_I(E_I(I)), P_T(E_T(y)) \rangle \tag{1}$$

The prediction class is then chosen greedily over all target classes $Y$ scaled by an optional temperature, $t$, $\arg\max_{y \in Y} \frac{S_y}{t}$.

### 3.1 SPURIOUS CORRELATION

We consider a dataset, $D$ consisting of $M$ samples with each sample represented as a tuple: $\{I, y^*, s\}$. $I$ is the input image, $y^* \in Y$ is the correct class, and $s \in S$ is the spurious attribute. In Waterbirds (Sagawa et al., 2019), $s$ is an attribute describing "*background*", or "*gender*" in datasets with gender bias. Previous studies have shown that zero-shot CLIP models are susceptible to spurious correlations, often associating a particular $s$ with a target class $y$, due to the imbalanced nature of the training distributions. We first define two sub-groups, $G_P$ and $G_N$ representing positive and negative associations, respectively. $G_P$ contain samples where the model infers a positive spurious relationship between $s$ and $y$, i.e. "*water background*" with "*waterbird*", while $G_N$ contains mismatched pairs like "*land background*". Models typically induce lesser errors on $G_P$ than on $G_N$, thus the goal is to reduce the performance gap, $G_P - G_N$, and improve $G_N$.

### 3.2 LINEAR DECOMPOSITION OF IMAGE REPRESENTATION

A recent work by Gandelsman et al. (2023) demonstrates that image representations in ViT can be expressed as a linear sum of contributions from individual layers. A ViT consists of $L$ layers, each made up of a Multi-Head Self-Attention (MSA) module with $H$ heads and a MLP module. The input image is first split into $N$ patches and projected onto $N$ embedding tokens, $\{z_i^0\}_{i,...N} \in \mathbb{R}^{N \times d}$, and prepended with a [CLS] token, $z_c^0$, here 0 refers to the embedding layer. We leave out the sample notation for brevity. The mechanistic framework of the ViT can be regarded as a residual stream with information being added at each layer. (Elhage et al., 2021) (see Fig. 1). We will focus on $z_c$ since the final prediction depends on that. We refer to the intermediate activations after each layer as **states**. Starting from $z^0$, we derive $L$ intermediate states:

$$\hat{z}^l = MSA^l(z^{l-1}) + z^{l-1}, \; z^l = MLP^l(\hat{z}^l) + \hat{z}^l., \; l \in L \tag{2}$$

The overall computations of $E_I(I)$ are then factorized as:

$$E_I(I) = z^0 + \sum_{l=1}^{L} MSA^l(z^{l-1}) + \sum_{l=1}^{L} MLP^l(\hat{z}^l). \tag{3}$$

Eq. 3 shows that we can decompose the final output of the image encoder into a linear sum of direct effects and similarly across each head and token in the MSA (Elhage et al., 2021; Gandelsman et al., 2023):

$$MSA^l(z^{l-1}) = \sum_{h=1}^{H} \sum_{i=0}^{N} \tilde{z}_i^{l,h}, \; \tilde{z}_i^{l,h} = a_i^{l,h} W_{V,O}^{l,h} z_i^{l-1}. \tag{4}$$

Here, $a_i^{l,h}$ and $W_{V,O}^{l,h}$ refer to the softmax attention weights and the combined value-output weight matrices of layer $l$ and head, $h$, respectively. Our work is partly inspired by Gandelsman et al. (2023), who showed that individual attention heads can be grounded onto natural language statements. This aligns well with the *"linear representation hypothesis"* (Elhage et al., 2022), which suggests that high-level concepts are linearly separable. We provide several empirical evidences of the concept separability in Sect. 6. Building on these ideas, we propose a contrastive approach that eliminates the need for manual interpretation, enabling more efficient and conclusive identification of attention heads for debiasing.

## 4 METHODOLOGY

We focus on intermediate states from attention heads aggregated over token positions, $\hat{z}^{l,h} = \sum_{i=0}^{N} \hat{z}_i^{l,h}$ and omit MLP layers from our study due to their limited direct impact (Gandelsman et al., 2023) and granularity. We introduce a method to detect salient attention states with high direct effect in Sec. 4.1, followed by our contrastive approach on locating class-discriminative and spurious states in Sec. 4.2 and then the debiasing techniques in Sec. 4.3.

### 4.1 LOCATING IMPORTANT ATTENTION STATES

We start by locating salient states that contribute significantly towards a target class. We utilize *Logit Lens (LL)* (Nostalgebrait, 2020), an interpretability technique that projects an intermediate state onto

the unembedding matrix. $LL$ enables visualizing the independent outcome of the intermediate state towards a target class. In CLIP, this is equivalent to decomposing the prediction logit into individual contributions by replacing the image representation in Eq. 1 with $\hat{z}^{l,h}$:

$$LL(l, h, y) = \langle P_I(\hat{z}^{l,h}), P_T(E_T(y)) \rangle. \tag{5}$$

We formulate the **direct effect** $v^{l,h}$ of an attention state towards a target class $y$ over the other class $\overline{y} \neq y$ as:

$$v^{l,h} = LL(l, h, y) - LL(l, h, \overline{y}). \tag{6}$$

In binary cases, $y$ represents the predicted class $\hat{y}$, with $\overline{y}$ as the other class, or the second most probable class in multi-class scenarios. By repeating Eq. 6 across all attention heads, we obtain a matrix $V \in \mathbb{R}^{L \times H}$, where each element captures the contribution of a state towards $y$ over $\overline{y}$. However, this would cause $V$ to have multiple low non-zero entries. We instead represent $V$ for each sample as a one-hot encoding:

$$v^{l,h} = \begin{cases} 1, & \text{if } (l,h) = \arg\max_{(l,h)} V, \\ 0, & \text{otherwise.} \end{cases} \tag{7}$$

In practice, $V$ is averaged over $D$ and normalized such that the direct effect across all heads sum to 1. We find that Eq. 7 enables $V$ to be a sparse matrix with $K \ll L \times H$ non-zero entries since earlier states tend to have lower direct effects (see C), analogous to Nostalgebraist (2020). We denote $P^*$ as the set of positions corresponding to the non-zero entries, $|P^*| = K$.

## 4.2 LOCATING SPURIOUS AND TARGET HEADS

**Spurious and Target states.** Additionally, we separately model the overall direct effects $V$ as an additive decomposition over components encoding the target $V_Y$ and spurious $V_S$ concepts:

$$V = V_Y + \alpha_{sy} V_S + \epsilon, \qquad V_Y = \sum_{i,j \in P_Y} v^{i,j}, \quad V_S = \sum_{i',j' \in P_S} v^{i',j'}, \quad \alpha_{sy} \in [-1, 1] \tag{8}$$

Here, $P_Y, P_S \subseteq P^*$ index attention heads whose corresponding direct effects $v^{i,j}$ predominantly support the target signal $Y$ or the spurious signal $S$, respectively. The coefficient $\alpha_{sy} \in \{-1, 1\}$ encodes the sign of the $S$–$Y$ association ($-1$ if $\{s, y^*\} \in G_N$, and 1 for $G_P$). The residual $\epsilon$ aggregates small contributions from MLP modules and head positions outside $P^*$. Empirically, we also observe a distinct *association* component between $Y$ and $S$; this is analyzed in Sec. 6. The definitions of $V_Y$ and $V_S$ are introduced below.

**Definition 4.1.** $V_Y$ represents the direct contribution of the concept $Y$ and $V_S$ of $S$ towards predicting $\hat{y} = y^*$ over $\overline{y} \neq y^*$, such that the following behavior can be observed:

$$\mathbb{E}_{V_Y \sim G_N}(V_Y | \hat{y} = y^*, a_{sy} = -1) > 0.$$
$$\mathbb{E}_{V_S \sim G_N}(V_S | \hat{y} = y^*, a_{sy} = -1) < 0. \tag{9}$$

Def. 4.1 states that when $\hat{y}$ is correctly predicted as $y^*$, the expected direct effect of $Y$ is positive and that of direct effects pertaining to $S$. The opposite is true when $\hat{y} \neq y^*$. This implies that $V_S$ negatively impacts the prediction of the correct class for samples within $G_N$, with $a_{sy} = -1$, i.e. *"land background"* is negatively associated with the true class of *"waterbird"*. We further divide $G_P$ into $\{G_{PW}, G_{PC}\}$ and $G_N$ into $\{G_{NW}, G_{NC}\}$, where $W$ and $C$ denotes wrongly and correctly classified subsets, respectively. We ignore the positively associated groups $G_P$, as both the spurious and target signals can co-occur ($\alpha_{sy} = 1$) and in this case, $V$ conflates $V_Y$ and $V_S$. Instead, we focus on the negatively associated groups $G_N$, where $\alpha_{sy} = -1$ induces opposing contributions from $V_Y$ and $V_S$, making each component easier to isolate.

**Spurious and Target contributions.** We formulate the contrastive solution to isolate $V_S$ as:

$$V_S = \sigma(V_{NW} - V_{NC}), \tag{10}$$

where $V_{NW}$ and $V_{NC}$ refer to computing $V$ with Eq. 6 and 7 by replacing $D$ with $G_{NW}$ and $G_{NC}$, respectively and setting $y = y^*$. The mask $\sigma(V) = \mathbb{1}(V > 0)$ filters out attention states with negative contributions. In Eq. 6, by definition $V_{NW} < 0$ for misclassified samples with $y = y^* \neq \hat{y}$

and $V_{NC} > 0$ for correct samples, $y = y^* = \hat{y}$. Under the decomposition in Eq. 8 and with $\alpha_{sy} = -1$), the direct effects $V$ is viewed as $V_Y - V_S + \epsilon$, leading to $V_{NW} = V_Y - V_S + \epsilon < 0$ and $V_{NC} = V_Y - V_S + \epsilon > 0$. Assuming the residual $\epsilon$ is small, these inequalities imply

$$V_{S|G_{NW}} > V_{Y|G_{NW}}, \qquad V_{Y|G_{NC}} > V_{S|G_{NC}}.$$

After normalizing each sample's contributions to unit mass, we obtain the monotone relations

$$\mathbb{E}[V_{S|NW}] > \mathbb{E}[V_{S|NC}], \qquad \mathbb{E}[V_{Y|NW}] < \mathbb{E}[V_{Y|NC}].$$

Consequently, the contrast in Eq. 10 thus only retain the positive $V_S$ contributions while masking out the negative $V_Y$ terms. By symmetry, swapping the order in Eq. 10 recovers the target $V_Y$ contributions.

**Intuition.** The key intuition is that the model's susceptibility to spurious attributes is more pronounced in incorrect predictions with $a_{sy} = -1$. When the model makes correct predictions under negatively spurious conditions, it is likely due to the influence of $Y$-relevant states outweighing the contribution of $S$-states (Def. 4.1), i.e. the representation of *"waterbird"* is stronger than the spurious cue, *"land background"* which leads to the correct prediction of *"waterbird"*.

However, we find that using a threshold of $0$ is not robust to noisy contributions that do not encode either $Y$ or $S$. To better isolate the dominant signals, we instead apply a mask $\sigma = \mathbb{1}(V > \gamma)$, with $\gamma = \frac{1}{|P^*_{|G_{NW}} \cup P^*_{|G_{NC}}|}$, where $P^*_{|G_{NW}}$ and $P^*_{|G_{NC}}$ are derived from the respective sub-groups. Additionally, we observe that targeting only the top state alone under this criterion already yields significant improvements (see Fig. 10 and B.3). Furthermore, one can sweep over $P_Y$ and $P_S$ after applying Eq. 10 on $G_{NC}$ and $G_{NW}$, respectively, to tune their selection on a validation set. In practice, this incurs little overhead due to the sparsity over $P_Y$ and $P_S$ induced by $\sigma$.

### 4.3 DE-BIASING ATTENTION STATES

This section discusses strategies to reduce spurious associations in CLIP and enhance performance in the worst-performing groups, i.e., $G_N$. As demonstrated in Sec. 4.2, states encoding $S$ can be identified, as they act as adversarial effects in $G_N$.

**Spurious ablation.** A straightforward solution is to eliminate these effects from the identified states. We use *mean-ablation (MA)* (Nanda et al., 2023) by setting each attention state in $Z_S$ to the mean value over the dataset, $\hat{z_S}^{l,h} = \frac{1}{M}\sum_{i=1}^{M} \hat{z}_{S,i}^{l,h}$. We did not find any difference between mean and zero ablation.

---

**Algorithm 1** Locate-Then-Correct

1: **Input:** Decomposed states $\{\hat{z}_i\}_{i=1}^{M}$, class positions $P_Y$, spurious positions $P_S$, class vectors $\{u_i\}_{i=1}^{N_u}$
2: **for** $(l, h) \in P_S$ **do**
3: $\quad \hat{z}^{l,h} \leftarrow \frac{1}{M}\sum_i \hat{z}_i^{l,h}$
4: **end for**
5: **for** $i = 1$ to $N_u$ **do**
6: $\quad$ **for** $(l, h) \in P_Y$ **do**
7: $\quad\quad \hat{z}^{l,h} \leftarrow \hat{z}^{l,h} + u_i \frac{\langle \hat{z}^{l,h}, u_i \rangle}{\langle u_i, u_i \rangle}$
8: $\quad$ **end for**
9: **end for**
10: **Return:** Debiased states $\{\hat{z}_i\}_{i=1}^{M}$

---

**Knowledge injection on target states.** To further enhance the class-discriminative properties of the identified class states $Z_Y$, we leverage LLMs, which have demonstrated significant potential in generating class-discriminative features through prompting (Adila et al., 2023; Menon & Vondrick, 2022; Yang et al., 2023b). We follow the same strategy in Adila et al. (2023) and prompt GPT4-o [1] to generate $N_u$ text features of each class, using a prompt, i.e., *"List the visual differences between waterbirds and landbirds"*. This gives us text insights, $s_y, s_{\overline{y}}$ (i.e. *"water background", "land background"*). The discriminative vectors are then obtained as $\{u_i\}_{i=1}^{N_u}$ by taking the normalized difference: $u = f_T(s_y) - f_T(s_{\overline{y}})/||f_T(s_y) - f_T(s_{\overline{y}})||$, where $f_T = P_T(E_T)$.

The selected attention states are then projected onto the discriminative vectors (Adila et al., 2023) before being added back, a process we refer to as *Knowledge Injection (KI)*: $z = z + u_i\langle z, u_i \rangle / \langle u_i, u_i \rangle$. The debiased states are then aggregated to form $E_I(I)$. Existing works (Adila et al., 2023; Chuang

---

[1] https://openai.com/index/hello-gpt-4o/

Table 1: Results on background bias. **Bolded** represents the best method while underline refers to the second best. **Metrics**: WG (↑), Avg (↑), Gap (↓)

| | ViT-B/16 | | | ViT-L/14 | | | ViT-H/14 | | |
|---|---|---|---|---|---|---|---|---|---|
| **Method** | WG | Avg | Gap | WG | Avg | Gap | WG | Avg | Gap |
| **Binary Dataset: Waterbirds** | | | | | | | | | |
| *Non-Parameter-tuning methods* | | | | | | | | | |
| ZS | 49.7 | 72.1 | 22.4 | 44.5 | 72.3 | 27.8 | 50.3 | 69.5 | 19.2 |
| TextSpan | 62.3 | **76.7** | 14.4 | 61.8 | 78.5 | 16.7 | 58.7 | 71.5 | 12.8 |
| Ortho-Cali | 68.1 | 73.3 | 5.2 | 73.3 | 78.6 | **5.3** | 18.1 | 41.9 | 23.8 |
| Roboshot | 57.5 | 63 | 5.5 | 70.5 | 79.0 | 8.5 | 60.7 | 71.6 | 8.9 |
| LTC (Ours) | 61.8 | 72.0 | 10.2 | **75.5** | **84.0** | 8.5 | 73.7 | 77.4 | 3.7 |
| LTC*(Ours) | **73.3** | 74.6 | **1.3** | 75.5 | 84.0 | 8.5 | **77.4** | **80.4** | **3.0** |
| *Parameter-tuning methods* | | | | | | | | | |
| ERM Probe | 35.2 | 78.5 | 43.3 | 62.5 | 86.3 | 23.9 | 56.2 | 85.2 | 29.0 |
| Cont Adapter | 83.2 | 86.9 | **3.7** | 83.7 | 89.3 | 5.6 | 87.5 | 91.3 | 3.8 |
| JTT | 72.3 | 87.2 | 14.9 | 81.6 | 90.9 | 9.3 | 85.7 | 91.4 | 5.8 |
| JTT-LTC (OURS) | **86.4** | **91.2** | 4.8 | **89.6** | **92.6** | 3.0 | **89.1** | **92.8** | 3.7 |
| **Multi-class Dataset: CounterAnimal** | | | | | | | | | |
| ZS | - | 54.8 | 21.1 | - | 66.1 | 15.8 | - | 72.7 | 14.7 |
| TextSpan | - | 52.6 | 23.3 | - | 62.5 | 19.4 | - | 71.4 | 16.0 |
| Ortho-Cali | - | 52.9 | 22.9 | - | 60.0 | 21.9 | - | 70.4 | 17.0 |
| Roboshot | - | 53.5 | 22.3 | - | 65.8 | 16.1 | - | 72.1 | 15.3 |
| LTC (OURS) | - | **55.2** | **20.7** | - | **66.3** | 15.6 | - | **73.8** | 13.6 |

et al., 2023) perform the projections at either the overall text $E_T(y)$ or image $E_I(I)$ representation space, whereas we propose to do so on specific attention states, hence the name of our framework, LOCATE-THEN-CORRECT. We detail the debiasing framework in Alg. 1. Note that MA and KI are sample-independent and applied in the same manner across the full inference dataset.

## 5 EXPERIMENTAL RESULTS

This section presents the empirical results of our debiasing framework on datasets exhibiting various spurious correlations. We focus on datasets where spurious correlation arises inherently in the model without parameter tuning, meaning the model does not develop bias toward spurious attributes due to imbalanced training data. Our analysis primarily addresses two types of biases: background-object class associations and gender-occupation correlations.

### 5.1 EXPERIMENTAL SETTING

**Dataset - Background Bias.** To evaluate robustness against background bias, we use the Waterbirds (WB) dataset (Sagawa et al., 2019), a binary classification task where zero-shot performance shows a significant gap between positive and negative subgroups. We report Worst-Group (**WG**), average accuracy (**Avg**), and the gap (**Gap**) between Avg and WG. We also consider a multi-class dataset: CounterAnimal (CA) dataset (Wang et al., 2024), which includes an easy subset ($D_E$) and a hard subset ($D_H$). We evaluate on $D_H$, using $D_E$ as a baseline. Since we are evaluating on $G_N$ itself, we re-use $Z_S$ from WB. Avg refers to the accuracy on $D_H$, and gap is between easy and hard set. Unlike the binary case in Waterbirds, the multi-class setting complicates the relationship between $y$ and $\overline{y}$, posing challenges for KI. To address this, we optimize the mapping of $y : \overline{y}$ for each class using $D_E$ and apply the same settings onto the baselines altogether, as discussed further in A.1.

**Dataset - Gender Bias.** GenderBias-VL (Xiao et al., 2024) consists of artificially generated images of working-class adults across 177 occupations, with both genders for each occupation. We study them across two tasks: occupation classification and image retrieval. For classification, we use the **Bias** metric, which measures the accuracy difference between gender groups: $\frac{1}{|O|} \sum_i^{|O|} |\text{Acc}(O_i | g = g_0) - \text{Acc}(O_i | g = g_1)|$, where $O$ represents occupations and $g_0, g_1$ denote male and female subgroups, respectively. We randomly select 25 occupations for optimization and evaluate on the remaining

Table 2: Results on Genderbias-VL - gender bias on occupation. $B_T$ refers to the 10 occupations with the highest discrepancy in classification between genders, chosen via zero-shot inference of the respective model. $B_O$ refers to bias across all occupations. $M_T$ and $M_O$ is the MaxSkew@10 of the top 10 and overall occupations. The objective is to achieve a low score across all metrics. Values ranges between 0 and 100.

| Method | ViT-B/16 | | | | ViT-L/14 | | | | ViT-H/14 | | | |
|--------|------|------|------|------|------|------|------|------|------|------|------|------|
| | $B_T$ | $B_O$ | $M_T$ | $M_O$ | $B_T$ | $B_O$ | $M_T$ | $M_O$ | $B_T$ | $B_O$ | $M_T$ | $M_O$ |
| ZS | 74.0 | 16.3 | 38.2 | 29.9 | 54.7 | 12.9 | 40.5 | 27.1 | 67.7 | 15.2 | 36.1 | 28.3 |
| TextSpan | 42.2 | 11.0 | 29.4 | 23.3 | 42.7 | 10.2 | 34.3 | 25.4 | 57.8 | 12.7 | 33.4 | 26.1 |
| Ortho-Cali | 36.2 | 10.2 | 39.3 | 25.7 | 29.7 | **9.2** | 36.8 | 25.8 | 32.5 | **9.4** | 33.1 | 22.7 |
| Roboshot | 65.2 | 18.3 | 38.3 | 30.1 | 52.6 | 14.0 | 40.0 | 29.1 | 65.1 | 16.4 | 35.4 | 28.3 |
| LTC (OURS) | **10.0** | **9.4** | **17.2** | **21.9** | **18.0** | 9.8 | **20.4** | **21.4** | **27.4** | 10.1 | **24.5** | **22.6** |

152. We select the top 10 biased occupations as $G_N$. We find that certain occupations are highly biased, and denote the top 10 occupations as WG. For retrieval task, we use the **MaxSkew@K** metric, defined as $\max_{g \in G} \log \frac{r_{g,k}}{1/|G|}$, where $r_{g,k}$ is the ratio of top $k$ images labeled with a specific gender. Additionally, we assess generalization performance on FairFace (Kärkkäinen & Joo, 2019) following the same settings as Chuang et al. (2023). While CelebA (Liu et al., 2015) is commonly studied for gender bias, we do not find significant bias present in zero-shot CLIP (Yang et al., 2023a) and leave it out of our evaluation. More details are provided in A.1.

**Baselines.** We evaluate on OpenCLIP (Ilharco et al., 2021) across 3 different sizes. We compare LTC against non-parameter tuning baselines including Zero-Shot **(ZS)**, **TextSpan** (Gandelsman et al., 2023), **Roboshot (RS)** (Adila et al., 2023) and **Ortho-Cali** (Chuang et al., 2023). Both Roboshot and Ortho-Cali perform debiasing via orthogonal projection on the overall input representation but differ on the modality: Image (Roboshot) and Text (Ortho-Cali). TextSpan constructs text representations per attention head and manually interprets them for debiasing. We follow the baseline implementations by not assuming access to group labels, and only use zero-shot predicted group labels. We denote **LTC**$^*$ when we optimized for $P_S$ and $P_Y$ given a validation set. While LTC is designed to address non-tuning methods, we find that it also serves as an simple and effective extension to parameter-tuning approaches. We apply LTC on top of the classifier trained with JTT (Liu et al., 2021), **(LTC-JTT)** and compare against **JTT**, Empirical risk minimization (ERM) probe **(ERM Probe)** and Contrastive Adapters **(Cont Adapter)** (Zhang & Ré, 2022). We use a 2-layer non-linear probe for all baselines except Cont Adapter, where we following the original settings in their work. Note that we only assume access to ground truth group labels for the validation set, following Cont Adapter. More implementation details are in A.2.

**Results - Background Bias.** As observed in Tab. 1, LTC surpasses all non-tuning baselines, bridging both the gap between groups and overall performance, similar findings can be observed in LTC-$\hat{S}$ with the exception of ViT-B. Despite being a lightweight extension to standard fine-tuning, JTT-LTC significantly outperforms both Cont Adapter and JTT, with fewer hyperparameters compared to Cont Adapter. In contrast, ERM exhibits strong bias toward majority groups. On CA, LTC is the only method that consistently improves over zero-shot CLIP, while others fail. We attribute Roboshot's failure to its reliance on LLMs for identifying spurious directions, which are often noisier than class-discriminative ones. Empirically, we find approaches that debias the overall representation are generally less effective than targeted interventions. Crucially, LTC's ability to localize contributions within $Z_Y$ and $Z_S$ improves interpretability (see Sec. 6). Ortho-Cali consistently underperforms, likely due to the complexity of solving a projection that equalizes representations across multiple spurious and target classes, which is challenging in multi-class settings.

**Results - Gender Bias.** Table 2 shows that zero-shot CLIP exhibits strong gender bias in certain occupations, with worst-group gaps significantly exceeding the average. In classification, Ortho-Cali performs comparably to LTC, except on ViT-B, where LTC clearly outperforms all baselines. In retrieval, TextSpan is competitive with LTC. Overall, LTC proves highly effective in mitigating gender bias across both classification and retrieval tasks. Similar to CA, we find that LLMs are often unreliable for identifying spurious features, as seen in Roboshot's failure—likely due to safeguards

that prevent the model from flagging gender as a potential source of bias. All 3 models predict $\hat{S}$ with 99% accuracy, with LTC-$\hat{S}$ yielding the same performance as LTC.

Tab. 3 shows the results of retrieving FairFace images pertaining to sensitive concepts, i.e., "*evil*", using the prompt: "*A photo of a [concept] person*" (Chuang et al., 2023), see Tab. 4. We apply the same optimization settings, i.e., $Z_S$ for LTC/TextSpan and text embeddings for Ortho-Cali from Genderbias-VL without further tuning. The evaluation, based on annotated gender attributes, measures MaxSkew@1000 and averaged across the concept set. The results demonstrate that LTC can automatically identify gender states similar to those found by TextSpan, without manual effort or external knowledge of the spurious information. This highlights the ability of our approach to filter attention states linked to spurious associations effectively.

## 5.2 Ablation Studies

Table 3: MaxSkew@1000 results on FairFace - Spurious relationship between gender and sensitive attributes. LTC (MA) only performs mean ablation on spurious states, without class-discriminative enhancement, as there are no target classes.

| Method | ViT-B/16 | ViT-L/14 | ViT-H/14 |
|---|---|---|---|
| ZS | 31.3 | 30.1 | 13.3 |
| TextSpan | **16.8** | 26.2 | 13.9 |
| Ortho-Cali | 23.8 | **21.4** | 17.8 |
| LTC (MA) | **16.8** | 26.2 | **12.0** |

In this section, we study the improvements from modular components of LTC. LTC (MA) only performs mean-ablation, without knowledge injection. The opposite is true for LTC (KI). LTC (R) performs both MA and KI on random states. We also include RS without removal of spurious features - RS (KI), which is equivalent to LTC (KI) but differs in where KI is applied. Tab. 6 illustrates the effectiveness of localized debiasing, with LTC (KI) performing substantially better. While MA improves over zero-shot settings, it proves to be insufficient compared to KI. The poor performance of LTC (R) highlights importance of correctly identified states. Overall, combining ablation and knowledge injection on optimally identified states yields the best results. Note that we do not perform any cross-validation to optimize $\gamma$, which we think may further improve the performance. Fig. 10 shows that LTC can locate the optimal states using as little as 20% of the labels. We provide more ablation studies, including robustness towards different text features in B.5 and B.3.

## 6 Interpreting Attention States

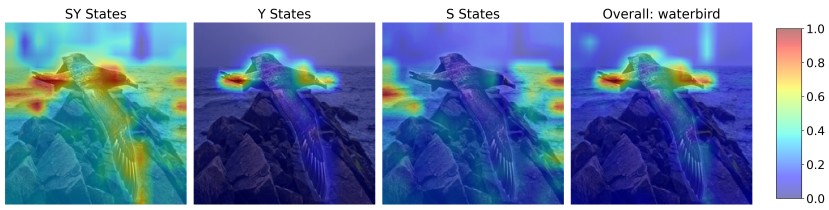

Figure 2: Image visualization: Localized representations of $Z_{SY}, Z_Y, Z_S$, and overall image.

**Spurious Association.** In Waterbirds, we found that there exist states encoding the knowledge of associating $S$ **with** $Y$, rather than $S$ alone, which we refer to as $SY$. To investigate this, we modify the task to classify the spurious attribute (background) instead of the target (bird) and change Eq. 10 to $\sigma(V_{NC})$. This locate contributions corresponding to classifying $S$ instead of $Y$. We first assume the new states as the actual $Z_S$ and the previous $Z_S$ as encoding the spurious association, $Z_{SY}$. Fig. 14 shows the highest contribution is located in a different state: L11H16, instead of L10H10 in Fig. 12. The left figure of Fig. 4 reveals that ablating $Z_S$ causes a much steeper drop in the classification of $S$ as compared to $Z_{SY}$. In TextSpan, the user is required to annotate the representation of each head and it is unclear if the heads encode $SY$ or $S$. Most heads are interpreted as $S$ and ablating them would erase the knowledge of $S$ directly.

Additionally, we study the effects of ablation on a more difficult task: predicting $S$ and $Y$ concurrently, i.e., "*A photo of a landbird on ocean*". As observed in the right figure of Fig. 4, the low performance drop indicates an important finding: $Z_{SY}$ represents knowledge of **associating** $S$ **with** $Y$ rather than $S$ **and** $Y$. We hypothesize the possibility behind this discovery as the contrastive approach finding reasons behind a wrong prediction, which in this case correctly refers to the model overly associating the background with the bird class. However, we do not find similar observations regarding gender bias, as $Z_S$ overlaps entirely with $Z_{SY}$. This may be due to background occupying a larger portion of the image, while gender is typically an intrinsic feature that occupies less visual space (see D.2). More importantly, the causal effects observed when ablating specific attention heads provide strong evidence that high-level concepts may be linearly separable across individual components.

**Text Interpretation.** TextSpan grounds visual representations onto generic text statements, which are not task-specific. Instead, we use GPT4-o to generate captions $c$ for each image and apply SHAP (Lundberg, 2017) to the identified states $Z_S, Z_{SY}, Z_Y$. SHAP assigns an importance score $\phi$ to each text token in the caption. The prediction logit is the similarity score between the sum over specific states and the text embedding: $\phi_r = \langle \sum_i^{|Z_r|} z_{r,i}, P_T(E_T(c)) \rangle \in \mathbb{R}^T$, where $r$ represents the encoded attribute type. The tokens are annotated by attribute ($S$ or $Y$), and the importance for each attribute is normalized over the caption length and averaged across tokens belonging within each attribute set. In Fig. 3, both $Z_Y$ and $Z_S$ allocates high importance to the attribute they represent and low importance to the other, while $Z_{SY}$ combines elements of both. This is aligned with Fig. 11, which lists the top text features for each state. $Z_Y$ mainly encoding species and various features of birds while $Z_S$ corresponds to habitat descriptions.

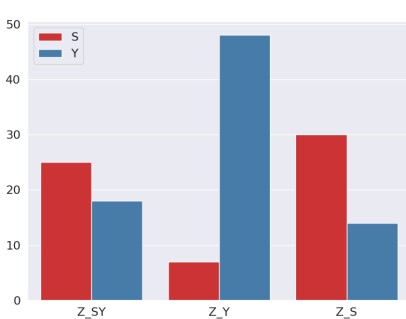

Figure 3: Normalized SHAP values for $Y$ and $S$.

**Image Interpretation.** We aggregate over the head and layer positions instead of the token positions in Eq. 4: $\sum_{(l,h)\in P_r} \tilde{z}^{l,h} \in \mathbb{R}^{N \times d}$ before deriving the prediction logit (Gandelsman et al., 2023). Fig 2 illustrates the magnitude of each pixel towards the prediction of "*landbird*" for $\{Z_{SY}, Z_Y, Z_S, P_I(E_I(I))\}$. Similar to text-based interpretations, $Z_{SY}$ shows high importance in patches corresponding to both the target class and background. In contrast, $Z_Y$ focuses more on the target class and less noisy as opposed to the overall representation, making it more effective for knowledge injection. We present more findings in D.

## 7 DISCUSSION

In this work, we propose our framework, Locate-Then-Correct (LTC) for debiasing CLIP models by identifying and correcting spurious associations in attention heads. By leveraging the linear decomposability of ViT representations, LTC enables fine-grained localization of attention states responsible for encoding spurious and target attributes. We found that implementing orthogonal projection on localized states yield superior results as compared to existing works which does so on the overall representation space. LTC, when used as a lightweight extension to existing fine-tuning methods, yields promising improvements. Furthermore, LTC provides an interpretable lens into the intermediate representations, enabling an explanation on why our debiasing measures work.

**Limitations.** A key limitation of our work is its exclusive focus on CLIP-based models. We believe that LTC is more broadly applicable. In particular, extending LTC to generative diffusion models could address bias amplification during image synthesis, where spurious cues often dominate generation. Similarly, adapting LTC to multimodal LLMs (e.g., LLaVA (Liu et al., 2023)) would allow debiasing in interactive settings where textual reasoning and visual grounding are tightly coupled. Exploring these directions would substantially broaden the impact of LTC and represents, in our view, an exciting and important avenue for future research.

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

## A EXPERIMENT INFORMATION

### A.1 DATASETS

**Waterbird:** Tab. 4 contains the details of the datasets used. In Waterbirds, the sizes of the validation set is skewed towards the positive set. In B.3, we show for certain model sizes, the convergence towards the optimal set of states can be achieved with a small sample size, $< 50$. We use the template: "A photo of a [class]" across all datasets.

Table 4: Dataset information: Val/Test size for $|G_N|$ and $|D|$. The sizes for the sub-groups in *GenderBias* are conditioned on the occupations.

| Dataset | $Y$ | $S$ | $G_N$ | $|G_N|$ | $|D|$ | License |
|---|---|---|---|---|---|---|
| Waterbirds | {landbird, water-bird} | {land, water} | {landbird in water, waterbird in land} | 240 / 2897 | 4795 / 5794 | MIT license |
| CounterAnimal | 45 ImageNet classes | {Snow vs Grass, Green vs Blue, ...} | {Polar bear on snow, White Brambling, ...} | – / 5926 | 7408 / 5926 | Unavailable |
| GenderBias | 177 Female/Male stereotypes | {female, male} | 10 worst occupations | – | 814 / 5093 | CC BY-NC-4.0 |
| FairFace | {good, evil, smart, dumb, attractive, unattractive, lawful, criminal, friendly, unfriendly} | {female, male} | – | – | 10954 | CC BY 4.0 |

**CounterAnimal.**   The dataset is divided into two subsets: 'easy' and 'hard'. The 'easy' subset contains examples of animals in their commonly observed habitats, while the 'hard' subset includes examples in rare or atypical habitats. For instance, a "*polar bear on snow*" represents an 'easy' example, whereas a "*polar bear on grass*" constitutes a 'hard' example. The objective is to minimize the classification gap between these subsets. The full dataset contains 45 ImageNet classes, however, we set $Y$ to the full Imagenet classes, $|Y| = 1000$. The multi-class nature of this task introduces a unique challenge: determining the appropriate counterfactual label $\overline{y}$ for a given class label $y$. Unlike binary classification, where a single $y, \overline{y}$ pairing suffices for KI, multi-class tasks involve multiple pairings for each class, and it is unclear which pairing to use for each image without prior knowledge of its class.

To address this issue, we first construct a dictionary for the 45 classes by identifying the most frequently misclassified class for each target. For example, if the class "*polar bear*" is frequently misclassified as "*arctic fox*", this pairing is recorded in the dictionary. This is recorded with the counts, generating a nested dictionary, with keys pointing to the CA classes and inner dictionaries corresponding to misclassified ImageNet classes. Next, we generate pseudo-labels, $y^p$, using zero-shot predictions to reference the dictionary. If $y^p$ corresponds to either the key or value in the dictionary, we retrieve the text features associated with that pair. To limit the possible text feature pairings which can be large, we pair each of the CA classes (outer key) to the misclassified class with the highest counts. Thus, each pseudo-label correspond to one of the CA classes, used to retrieve the text features. However, this introduces a limitation in the event where the pairing between the pseudo label and CA class does not correspond to the actual text feature pairing, ie pseudo-label of *"seal"* to CA class of *"polar bear"* but text pairing is *"polar bear - arctic fox"*. However, we find that this can still introduce benefits by endowing the model with knowledge of discriminatory features related to the CA class. Though it is possible to generate text features for all pairings, we leave investigation

of this to future works. It is important to note that we do not limit the predictions to only the CA classes, but only retrieve text features limited to them. The total classes as normalized over by the classifier is still the full ImageNet classes. Overall, this process can be interpreted as a refinement stage: an initial prediction is made, followed by the injection of discriminative features to improve classification accuracy. This methodology is similarly applied to RoboShot and Ortho-Cali.

**GenderBias.** In the GenderBias dataset, each image is linked to a target occupation $y$, such as "*Pilot*", and is annotated as stereotypically male or female based on data from the U.S. Bureau of Labor Statistics (BLS)[2]. The alternative class, $\overline{y}$, represents an occupation stereotypically associated with the opposite gender, such as "*Flight Attendant*". All occupations in the dataset include samples from both genders, and the bias metric measures accuracy discrepancies between them. In the original dataset, each occupation is paired with multiple correlated occupations. We instead choose the occupation with the highest proportion of workers from the opposite gender. For example, for *Flight Attendant*, *Pilot* is chosen over *Airline Manager* if it has a higher male labor force representation. Certain occupations exhibit stronger gender bias in CLIP. To simulate $G_N$, we select the 10 occupations with the highest bias scores during zero-shot inference. Across the three models analyzed, consistent patterns emerge, with occupations such as "*Lawyer*" and "*Chief Executive*" being strongly associated with males. During optimization over $P_Y$ and $P_S$, we use the top 10 occupations with highest bias within the validation set.

**FairFace.** FairFace comprises images of occupation-neutral faces annotated with gender. Following the settings in (Chuang et al., 2023), we prompt CLIP to retrieve the top $K$ images associated with each concept class in $Y$. MaxSkew quantifies the maximum gender skewness across all concepts in $Y$ and averages these values. The target and spurious heads identified in GenderBias are reused in FairFace without further optimization or reliance on a validation set.

## A.2 BASELINES

**TextSpan (Gandelsman et al., 2023).** The framework maps the representation of each attention head in a ViT model to a set of natural language text statements generated using ChatGPT. Users then determine whether a head should be categorized as $Y, S$ or neither based on these statements. For example, a set of statements like "*Submerged underwater scene, Peaceful rural farmland, ...*" might be labeled as $S$ for Waterbirds. However, this approach is subject to individual interpretation, potentially leading to disagreements among evaluators. Additionally, the manual effort required increases significantly as the number of attention heads grows. TextSpan is implemented on ImageNet (Deng et al., 2009) and does not utilize the validation set of the benchmarks.

**Ortho-Cali (Chuang et al., 2023).** This approach leverages *positive pairs* of text prompts, enforcing the projection matrix to regularize the difference between two projected embeddings with opposing spurious attributes. The pairs are structured as "a photo of a [class name] with [spurious attribute]." Consequently, the method requires prior knowledge of the spuriously correlated attribute, such as *male* and *female* for GenderBias. The projection matrix is derived from the validation set and applied on the **text representation** $P_T(E_T(y))$ during testing.

**Roboshot (Adila et al., 2023).** Roboshot uses an LLM to generate helpful and harmful concepts related to classifying $Y$. Harmful concepts are treated as $S$, while helpful concepts are used to enhance CLIP's discriminative ability. Harmful concepts are removed from the final **image representation** $P_I(E_I(I))$, and helpful concepts are amplified through orthogonal projection. As with Ortho-Cali, the projection matrix is derived from the validation set.

**Parameter-tuning baselines.** JTT (Liu et al., 2021) is a two-stage framework where an initial model is trained to identify the worst-performing examples, which are then emphasized in a second training stage by upsampling them with a factor, $\lambda_{up}$. However, given CLIP's strong performance in zero-shot settings, we omit the first stage and directly predict group labels to identify $G_N$ as the set of worst-performing samples. As with the non-training setup, JTT-LTC operates purely at inference time.

---

[2]https://www.bls.gov/cps/cpsaat11.htm

We first train the base JTT model, then apply mean ablation and knowledge injection to the identified attention states prior to aggregating them into the final image representation for prediction. Note that the group labels used to identify $Z_S$ and $Z_Y$ are inferred in a zero-shot fashion. During JTT training, we set $\lambda_{up} = 90$ for ViT-H/14 and 100 for ViT-B/16 and ViT-L/14. We use a learning rate of $1e-2$ and weight decay as $1e-4$. We adopt a 2-layer non-linear probe as the classifier for ERM, JTT and JTT-LTC, with the hidden layer dimension as 128 for ViT-B/16 and 256 for ViT-L/14 and ViT-H/14. Cont Adapter uses a 2-layer adapter instead. We ran the Cont Adapter with the original hyperparameters and only change the CLIP backbone.

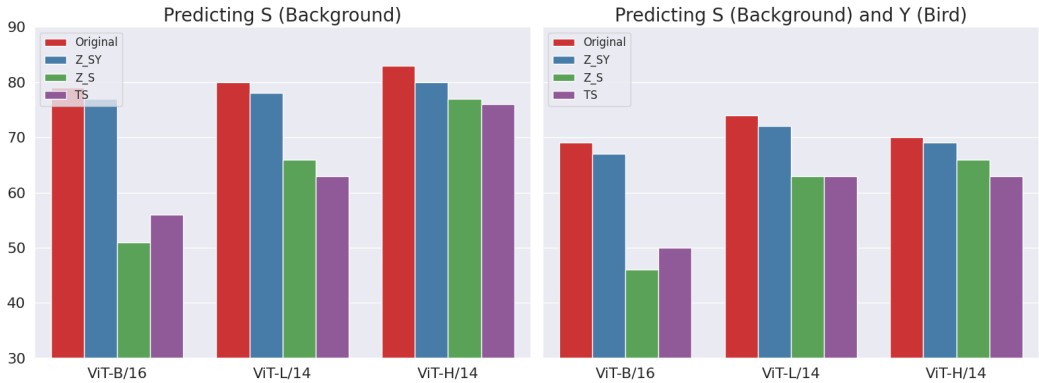

Figure 4: **Z_S**: Ablating states encoding $S$, **Z_SY**: Association between $S$ and $Y$. **TS:** TextSpan **[Left]:** Predicting $S$ as the target class. **[Right]:** Predicting both $S$ and $Y$. **Dataset: Waterbirds**

Table 5: Prompts to generate class discriminative concepts. (Adila et al., 2023). Replace *"visual"* with *"spurious"* for spurious concept.

| Dataset | Prompt |
|---|---|
| Waterbirds | List the true visual differences between waterbirds and landbirds. Give short keyword for each answer. Answer in the following format: <Difference>: <waterbird characteristic> ; <landbird characteristic> |
| Genderbias/CA | List 3 true visual differences between {cls1} and {cls2}. Give short keyword for each answer. Answer in the following format: <Difference>: <{cls1} characteristic> ; <{cls2} characteristic> |

## B ADDITIONAL RESULTS

### B.1 ANALYSIS ROBUSTNESS BETWEEN SUB-GROUPS

In this section, we analyze the distribution of prediction margins, $p(\hat{y}) - p(\overline{y})$, for $G_P$ and $G_N$. Examples of $G_P$ and $G_N$ can be referenced from Tab. 4. The results across the three models and four baselines are shown in Fig 5, 6 and 7. In the zero-shot setting, a clear separation between the sub-groups is observed, with $G_N$ skewed toward the negative end. Baselines designed to remove spurious correlations between $S$ and $Y$ often introduce a trade-off between sub-group accuracies, as the positively correlated spurious attribute may have contributed to better predictions for classes with positive $a_{sy}$. Among the three baselines, LTC uniquely avoids this trade-off. For the large and huge models, both $G_P$ and $G_N$ shift toward the positive margin, a trend more pronounced in the huge model. While Ortho-Cali and Roboshot improve performance on $G_N$, they compromise on $G_P$. Roboshot outperforms Ortho-Cali by amplifying helpful concepts but falls short of LTC, which achieves better results through head-level optimization.

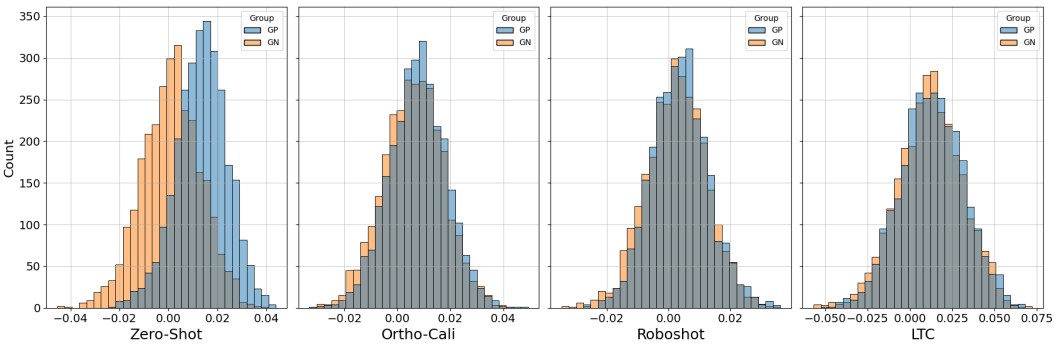

Figure 5: Prediction margins in Waterbirds. **Model: ViT-B/16**

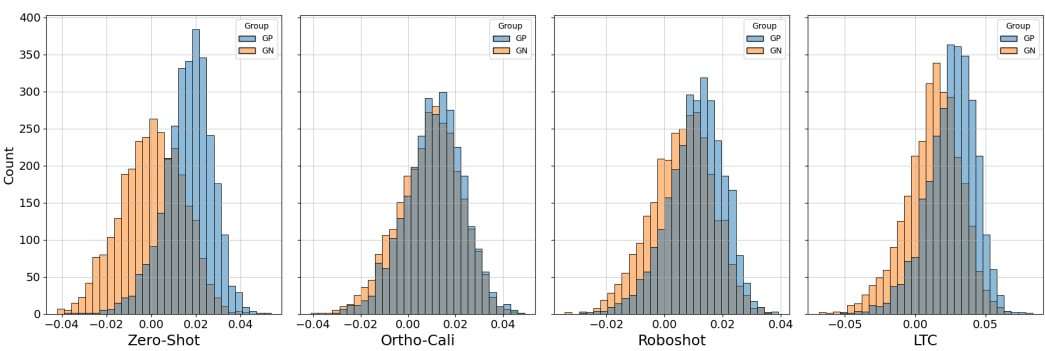

Figure 6: Prediction margins in Waterbirds. **Model: ViT-L/14**

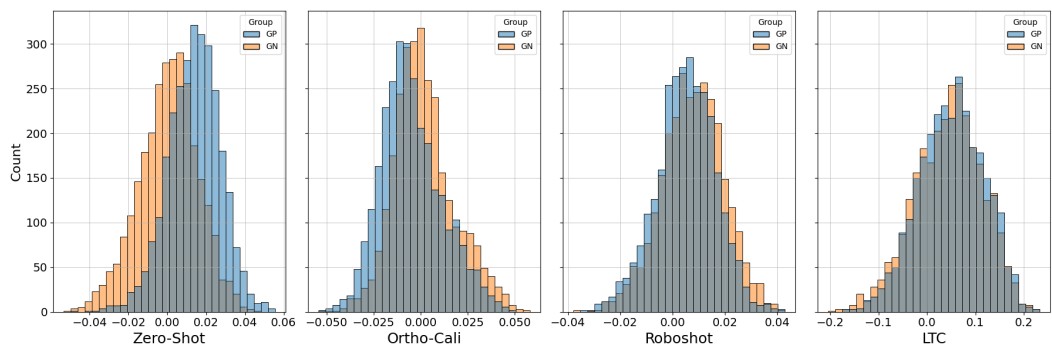

Figure 7: Prediction margins in Waterbirds. **Model: ViT-H/14**

Table 6: Ablation studies on Waterbirds. **LTC (MA)** - Ablate only, **LTC (KI)** - Knowledge injection only. **LTC (R)** - Similar to LTC but states are randomized. Roboshot (RS) (KI): Only KI without debiasing. Worst Group (**WG** ↑), **Gap** = Avg - WG (↓)

| Method | ViT-B/16 | | ViT-L/14 | | ViT-H/14 | |
|--------|------|------|------|------|------|------|
| | WG | Gap | WG | Gap | WG | Gap |
| ZS | 49.7 | 22.4 | 44.5 | 27.8 | 50.3 | 19.2 |
| RS | 57.5 | 5.5 | 70.5 | **8.5** | 60.7 | 8.9 |
| RS (KI) | 45.6 | 23.6 | 45.2 | 26.6 | 42.8 | 21.8 |
| LTC (MA) | 62.5 | 13.1 | 51.8 | 23.2 | 60.5 | 11.6 |
| LTC (KI) | 67.3 | 6.6 | 72.9 | 10.1 | 69.7 | 5.3 |
| LTC (R) | 36.9 | 35.7 | 15.2 | 42.4 | 43.1 | 11.3 |
| LTC | **73.3** | **1.3** | **74.6** | 9.6 | **71.3** | **3.6** |

## B.2 GENDERBIAS

**Ablation.** Tab. 7 presents the results of ablating various components of LTC for GenderBias. Overall, Roboshot underperforms compared to LTC and even increases the overall bias relative to zero-shot performance. Performing KI without orthogonalizing out spurious feature achieves a lower bias for RoboShot. As discussed in the main results, the reliance on LLM to identify spurious features may backfire if the LLM is sufficiently safeguarded against generating sensitive information such as gender being a prominent correlation to occupations. Similar trends to Waterbirds are observed, where LTC (KI) emerges as the second most competitive baseline. Despite using the same helpful concepts, LTC (KI) significantly outperforms RS (KI), demonstrating that orthogonal projection on classification heads is a more effective method for amplifying class-discriminative properties in CLIP.

**Occupations analysis.** Fig. 8 presents statistics on bias relative to male workforce proportions. Common biased occupations include *Chief Executive*, *Lawyer*, and *Security Guard*, all of which are male-dominated. A clear trend emerges: male-dominated occupations exhibit higher bias levels, while female-dominated occupations show lower bias. Additionally, occupations associated with the opposite gender tend to exhibit reduced bias. This suggests that CLIP is disproportionately influenced by gender bias across occupations. For example, while CLIP accurately classifies both male and female "*Legal Secretaries*", it demonstrates significantly higher accuracy for male "*Lawyers*" compared to female. The positive correlation between classification performance and workforce proportions indicates that CLIP is heavily impacted by the gender composition of occupations present in its training data.

Table 7: Ablation studies on **Genderbias**. **LTC (MA)** - Ablate only, **LTC (KI)** - Knowledge injection only. **LTC (R)** - Similar to LTC but states are randomized. Roboshot (RS) (KI): Only KI without debiasing. Worst Group Bias $B_{10}$ (↓), Overall Bias $B_{ovl}$(↓) **Bolded**: represent best method while underline refers to second best.

| Method | ViT-B/16 | | ViT-L/14 | | ViT-H/14 | |
|---|---|---|---|---|---|---|
| | $B_{10}$ | $B_{ovl}$ | $B_{10}$ | $B_{ovl}$ | $B_{10}$ | $B_{ovl}$ |
| ZS | 74.0 | 16.3 | 54.7 | 12.9 | 67.7 | 15.2 |
| RS | 65.2 | 18.3 | 52.6 | 14.0 | 65.1 | 16.4 |
| RS (KI) | 56.7 | 14.3 | 48.6 | 11.4 | 57.7 | 13.3 |
| LTC (MA) | 43.9 | 10.9 | 43.7 | 10.7 | 59.5 | 12.4 |
| LTC (KI) | 18.2 | 10.7 | 24.8 | 10.3 | 30.3 | 10.6 |
| LTC (R) | 30.5 | 12.2 | 35.3 | 14.6 | 33.4 | 15.0 |
| LTC | **10.0** | **9.4** | **18.0** | **9.8** | **27.4** | **10.1** |

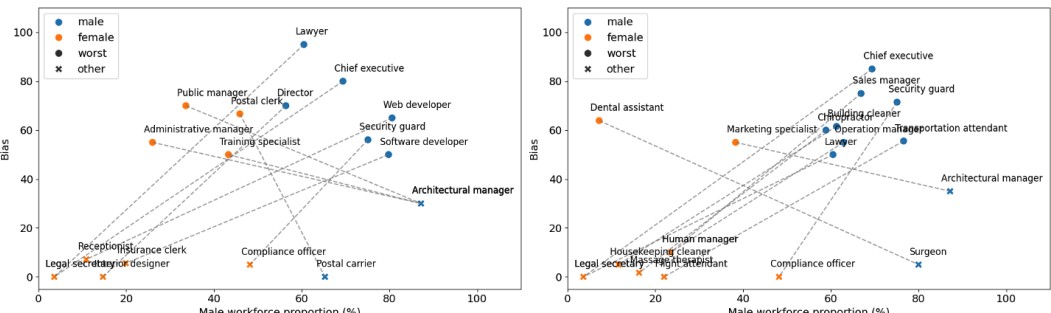

Figure 8: Qualitative analysis on gender bias of top 10 occupations vs proportion of male workforce. Worst: occupation, other: opposite-gender associated occupation. **Left: ViT-B/16**, **Right: ViT-H/14**

### B.3 SAMPLE SIZE AND MASK

In Sec. 5, we did not optimize for the optimal set of attention states to perform debiasing. Both $Z_Y$ and $Z_{SY}$ were chosen through the mask, $\sigma$, and filtering out states with contribution $< \gamma$. We find that this works well as a heuristic at filtering out noisy states which may not correspond to either $S$ or $Y$. Fig. 10 and 11 shows the ablation study on sample size for Waterbirds and Genderbias, respectively. Specifically, we observe the performance trend vs % of $G_P$ and $G_N$ utilized on the primary step of LTC: locating $Z_Y$ and $Z_{SY}$. We observe that the trend differs between model sizes. ViT-B peaks at the later stages, middle for ViT-H and early for ViT-L. We found that larger models tend to allocate higher contributions within a single $Z_{SY}$ (59%) and $Z_Y$ (58%), see Fig. 15, 18 for Waterbirds. More importantly, the important state contributions in $V_{NC}$ and $V_{NW}$ are not similar in magnitude, thus preventing being canceled out. On the other hand, ViT-B has higher overlapping values between the important states: the state at layer 11, head 5 is relatively high on both correct and wrong samples.

We additionally analyze the effects of selecting contributing states filtered with the mask, by only restricting $Z_Y$ and $Z_{SY}$ to the single top state. This essentially avoids using $\gamma$ as a threshold. We find that performing debiasing on the incomplete states tends to underperform, except for ViT-L, as selecting the top states essentially recovers the heuristic setting since $|Z_Y| = 2$ and $|Z_{SY}| = 1$. Overall, we find that using the mask effectively finds states most probable for encoding the respective representations.

### B.4 SPURIOUS EFFECTS IN NEGATIVE SUB-GROUP

The correct discovery and distinction between $Z_Y$ and $Z_S/Z_{SY}$ are hinged on the belief that the two can be separated more accurately in $G_N$. We study the implications of neglecting this belief by examining the located states when implementing Eq. 10 across the full dataset instead, replacing

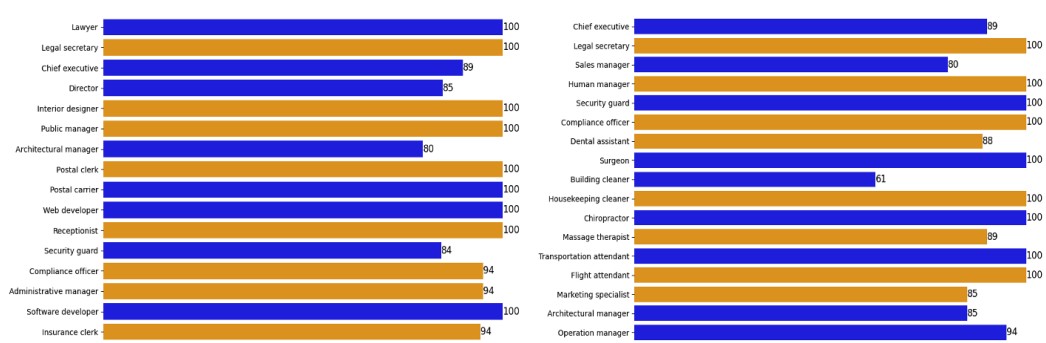

Figure 9: Accuracy of the occupation-dominated gender of each occupation. **Left: ViT-B/16**, **Right: ViT-H/14**

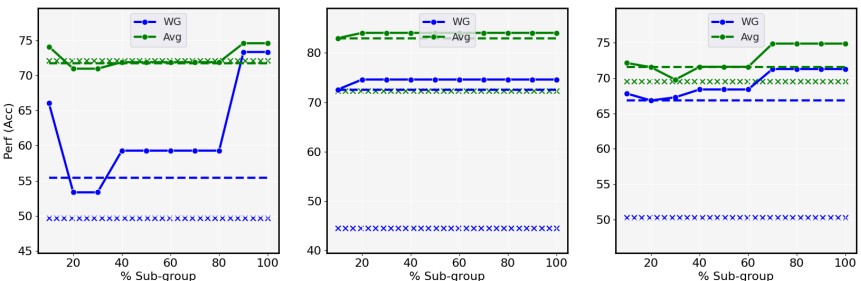

Figure 10: Analysis on % sub-group samples for locating spurious and classification states against performance. **Left: ViT-B/16**, **Middle: ViT-L/14 Right: ViT-H/14**. **Dash lines** refer to selecting the top contributing state in $Z_Y$ and $Z_{SY}$. **x** refers to ZS. **Metric: Accuracy (↑). Dataset: Waterbirds**

both $V_{NW}$ and $V_{NC}$ with $V_W$ and $V_C$, respectively. From Tab. 8, we can see that performing the locating framework on the full dataset in place of $GN$, causes a huge performance drop for ViT-B. The identified $Z_Y$ is confused with $Z_{SY}$, causing KI to be performed on the spurious states instead of class states. On ViT-L, the located states converges to the same set as in $G_N$, but we observe the contributions on the actual states to be lower. ViT-H appears to be similarly robust, but still suffers a marginal performance drop and lower $Y/S$ ratio in $Z_Y$.

## B.5 SENSITIVITY TO DISCRIMINATIVE FEATURES

Since KI depends on the quality of the underlying discriminative features, we assess its robustness to variations in the prompt set, $\{u_i\}_{i=1}^{N_u}$. We measure the mean and standard deviation between 3 sets of features, between RS and LTC. We report the mean and standard deviation over three feature sets for both RS and LTC. As shown in Tab. 9, LTC demonstrates greater robustness, likely due to its focus on states with high class-relevance, making it better suited for injecting discriminative information.

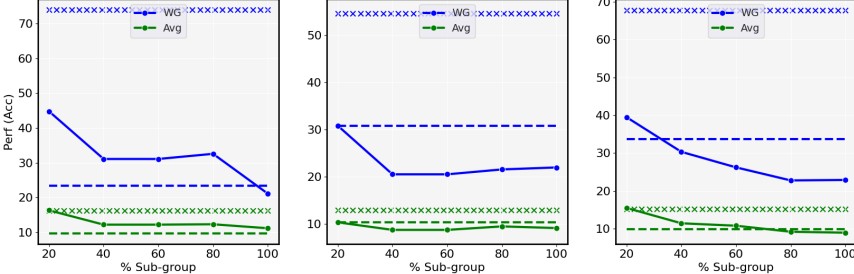

Figure 11: Analysis on % sub-group samples for locating spurious and classification states against performance. **Dash lines** refer to selecting the top contributing state in $Z_Y$ and $Z_{SY}$. **x** refers to ZS. **Metric: Bias ($\downarrow$). Dataset: Genderbias**

Table 8: Analysis of samples used for spurious and class state identification. The metric for $Z_{SY}$ and $Z_Y$ is the SHAP scores of $Y/S$ features. $Z_Y$ should have higher contributions on $Y$ as opposed to $S$. WG is the worst-group accuracy. ViT-L/14 is not shown as the states are identical. Negative: $G_N$, Full: $D$. **Dataset: Waterbirds**

| Method | ViT-B/16 | | | ViT-H/14 | | |
|---|---|---|---|---|---|---|
| | $Z_{SY}$ | $Z_Y$ | WG | $Z_{SY}$ | $Z_Y$ | WG |
| Full | 30/15 | 25/18 | 15 | 26/12 | 44/7 | 69.3 |
| Negative | 25/18 | 8/48 | 73.3 | 18/25 | 35/7 | 71.3 |

Table 9: Mean and std. performance on **Waterbirds** across 3 sets of discriminative features.

| Method | RS | | | LTC | | |
|---|---|---|---|---|---|---|
| | WG | Avg | Gap | WG | Avg | Gap |
| ViT-B/16 | $60.8 \pm 2.4$ | $67.2 \pm 3.0$ | $6.4 \pm 0.7$ | $73.6 \pm 0.5$ | $76.5 \pm 2.5$ | $3.0 \pm 2.0$ |
| ViT-L/16 | $68.8 \pm 4.2$ | $78.3 \pm 2.3$ | $10.3 \pm 2.1$ | $74.4 \pm 2.9$ | $83.9 \pm 1.3$ | $9.5 \pm 1.6$ |
| ViT-H/14 | $59.5 \pm 6.2$ | $73.4 \pm 1.3$ | $13.9 \pm 6.7$ | $66.7 \pm 5.5$ | $75.6 \pm 2.8$ | $8.9 \pm 4.1$ |

## C    CONTRIBUTION DISTRIBUTIONS

Fig. 12, 15 and 18 refers to the target class $Y$'s contribution distribution in Waterbirds. The top and bottom heatmaps correspond to samples in $G_{NC}$ and $G_{NW}$. Fig. 13, 16 and 19 similarly correspond to Genderbias. Fig. 14, Fig. 17 and Fig. 20 refers to correctly classified background $S$ samples in the entire dataset $D$.

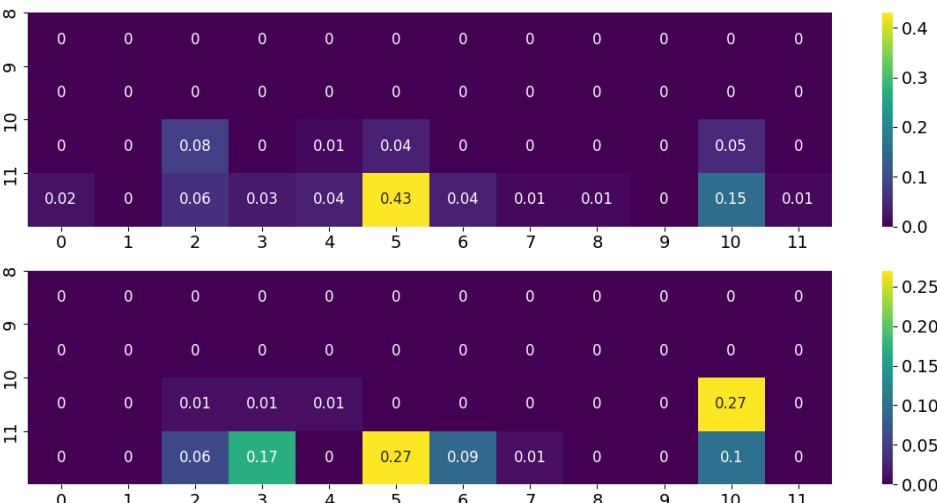

Figure 12: $V$ scores across the last 4 layers and all heads for **ViT-B/16**. Layer-wise in the y-axis and head-wise in x-axis. **Top:** $V_{NC}$, **Bottom:** $V_{NW}$. **Dataset: Waterbirds**

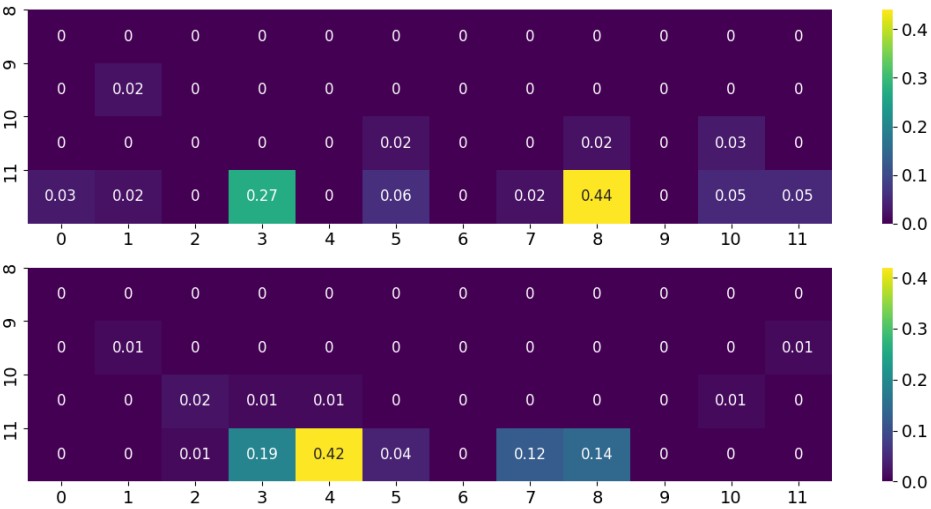

Figure 13: $V$ scores across the last 4 layers and all heads for **ViT-B/16**. Layer-wise in the y-axis and head-wise in x-axis. **Top:** $V_{NC}$, **Bottom:** $V_{NW}$. **Dataset: Genderbias**

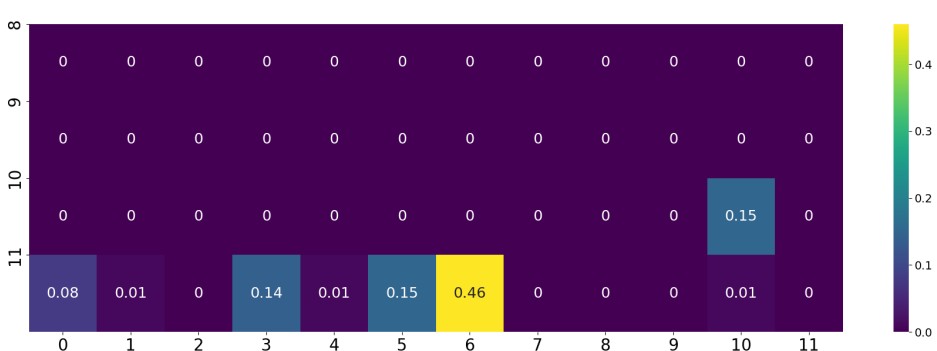

Figure 14: $V_C$ scores across the last 4 layers and all head for predicting the **spurious attribute: background** directly for **ViT-B/16**. Layer-wise in the y-axis and head-wise in x-axis. **Dataset: Waterbirds**

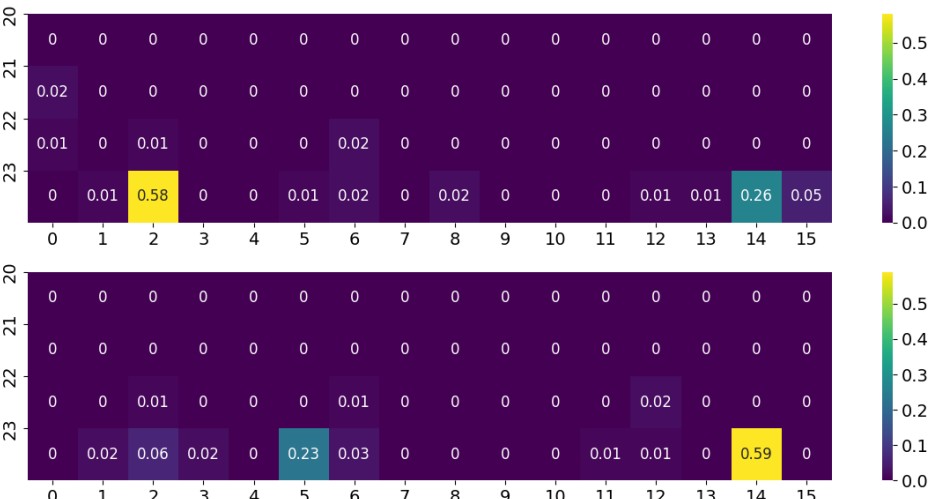

Figure 15: $V$ scores across the last 4 layers and all heads for **ViT-L/14**. Layer-wise in the y-axis and head-wise in x-axis. **Top:** $V_{NC}$, **Bottom:** $V_{NW}$. **Dataset: Waterbirds**

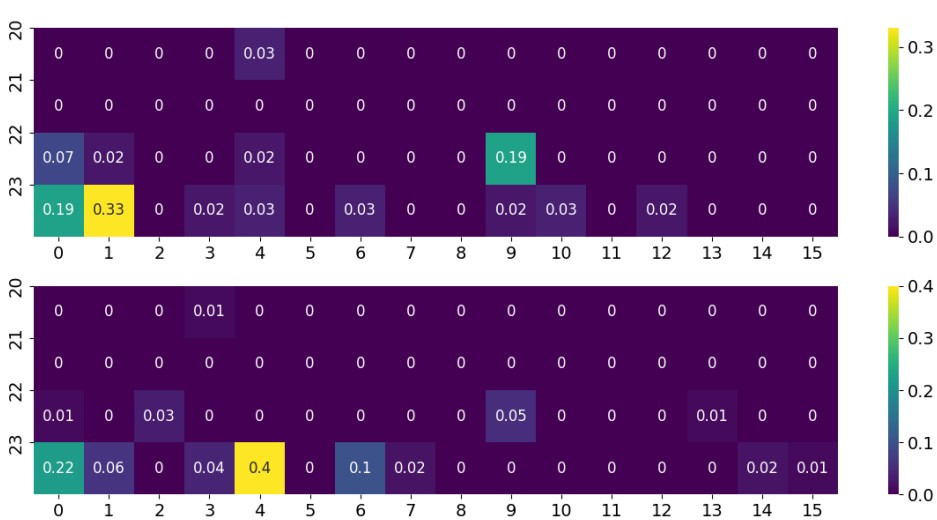

Figure 16: $V$ scores across the last 4 layers and all heads for **ViT-L/14**. Layer-wise in the y-axis and head-wise in x-axis. **Top:** $V_{NC}$, **Bottom:** $V_{NW}$. **Dataset: Genderbias**

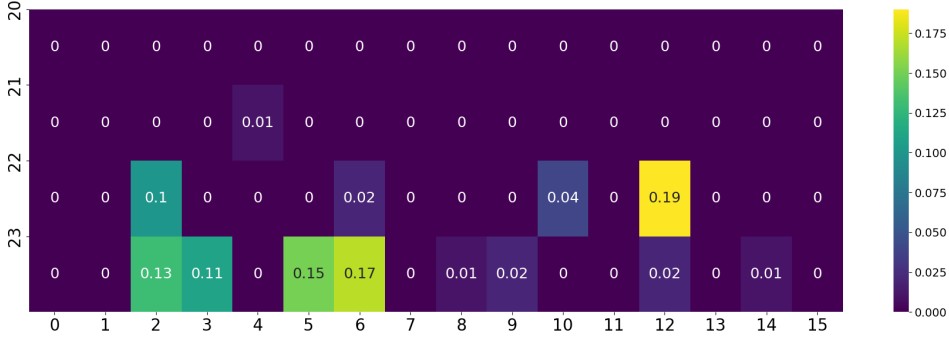

Figure 17: $V_C$ scores across the last 4 layers and all head for predicting the **spurious attribute: background** directly for **ViT-L/14**. Layer-wise in the y-axis and head-wise in x-axis. **Dataset: Waterbirds**

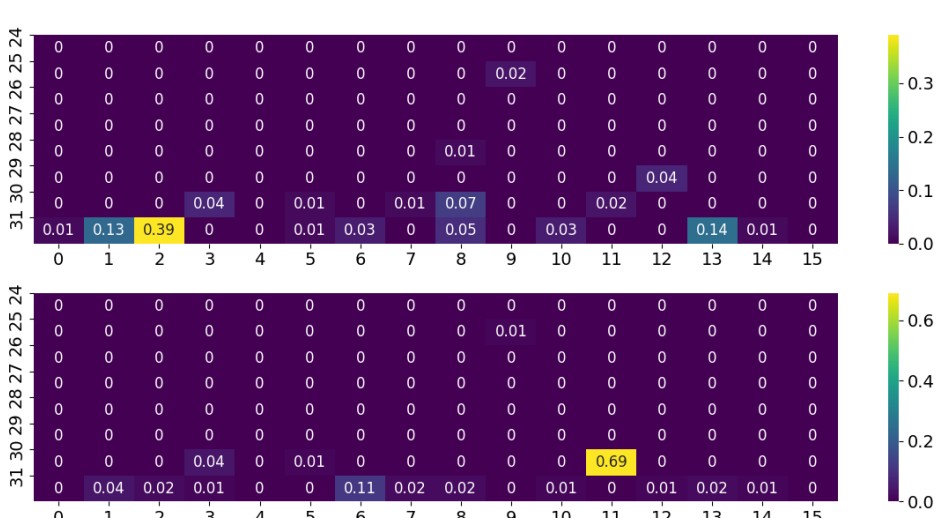

Figure 18: $V$ scores across the last 8 layers and all heads for **ViT-H/14**. Layer-wise in the y-axis and head-wise in x-axis. **Top:** $V_{NC}$, **Bottom:** $V_{NW}$. **Dataset: Waterbirds**

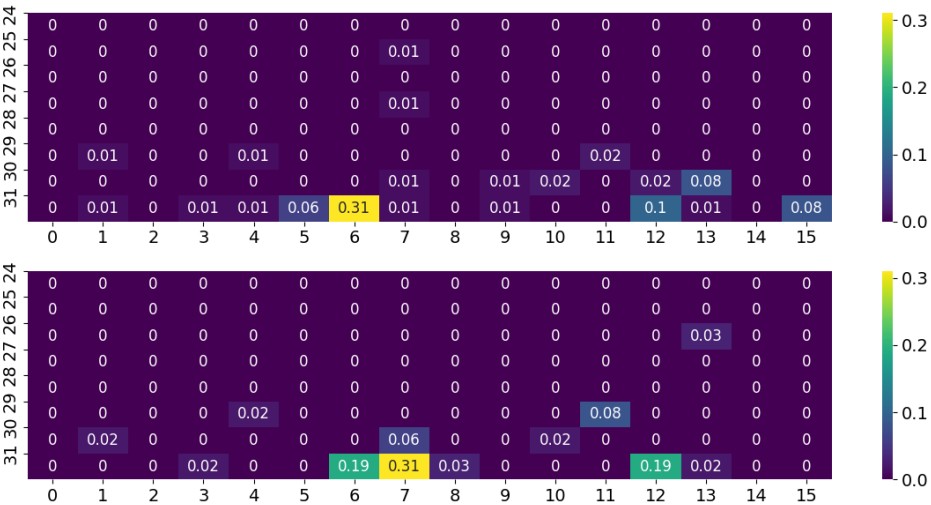

Figure 19: $V$ scores across the last 8 layers and all heads for **ViT-H/14**. Layer-wise in the y-axis and head-wise in x-axis. **Top:** $V_{NC}$, **Bottom:** $V_{NW}$. **Dataset: Genderbias**

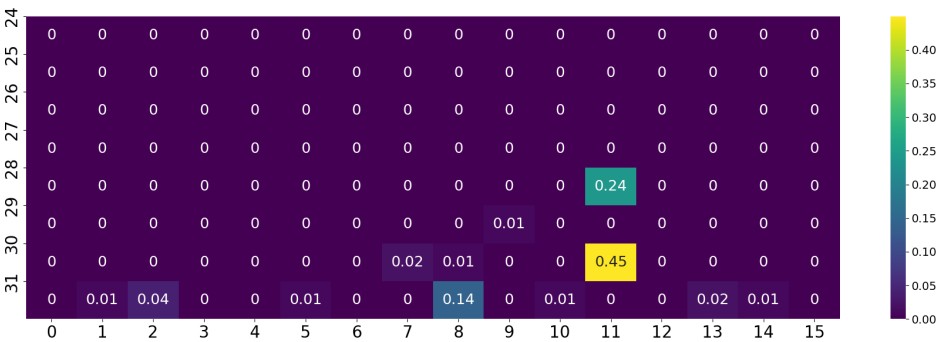

Figure 20: $V_C$ scores across the last 4 layers and all head for predicting the **spurious attribute: background** directly for **ViT-H/14**. Layer-wise in the y-axis and head-wise in x-axis. **Dataset: Waterbirds**

# D INTERPRETABILITY RESULTS

We present interpretability findings for each model in this section. To generate captions for SHAP, we prompt MiniCPM-V 2.6 (Hu et al., 2024) using the templates shown in Tab. 10. Fig. 21 shows the normalized SHAP values for the set of aggregated states representing $\{SY, S, Y\}$ for Waterbirds or $\{S, Y\}$ for Genderbias. Referencing the prompt, we form $Y$ from "*features*" and the annotated "*species*" of the bird for Waterbirds and the occupation class for Genderbias. To get the normalized SHAP score for $Z_Y$, we take the max over the $Y$ set, i.e., max over features or species for Waterbirds. $S$ is then "*background*" and "*gender*" for Waterbirds and Genderbias, respectively. We also provide additional SHAP scores for each individual state and attribute, namely, each state against each attribute, extracted from the captioning output. This provides a fine-grained analysis of the type of information that is encoded and aggregated across each attribute state. The top text features are retrieved by aggregating over the number of times a particular feature is ranked as the feature with the highest SHAP value in each sample.

Table 10: Prompts to caption each image using GPT4o.

| Dataset | Prompt |
| --- | --- |
| Waterbirds | Caption the picture of the {class} and describe both the visual features of the bird itself and the background. Please format your response as follows: Caption: Background: Features: |
| Genderbias | Caption the picture of the {occupation}. Describe the visual features in the picture that correlate with the occupation of {occupation} and the gender of the {occupation}.Please format your response as follows: Caption: Gender: Features: |

## D.1 WATERBIRDS

Fig. 21 illustrates the normalized SHAP scores for each feature category, corresponding to distinct attribute states. Our analysis reveals that localized states exhibit strong correlations with their hypothesized attributes and minimal correlations with opposing attributes. Specifically, $Z_Y$ assigns a high contribution to features associated with $Y$ while attributing minimal contribution to $S$. Similar patterns are observed in Tab. 11, where $Z_Y$ predominantly contains terms related to various bird species and their features, whereas $Z_S$ primarily represents the surrounding background.

**Correlation with TextSpan**. We applied TextSpan (Gandelsman et al., 2023) to each state and observed that the textual interpretations align with the intended state representations. On both ViT-B and ViT-H, the derived texts for $Z_{SY}$ correspond to background descriptions, whereas on ViT-L, they relate to object categories. Notably, this is consistent with the composition of SHAP values shown in Fig. 21. For $Z_Y$ and $Z_S$, the text outputs are fully aligned with their respective intended representations.

**Individual state representation.** Tab. 14 lists the top 10 text features for each individual head within the target set, $\in Z_Y$. Most heads predominantly represent the *species* category. However, in ViT-H/14, we identified a specific head, L31H13, that focuses on representing various colors and has a higher contribution to the *features* category. Similarly, L31H1 exhibits comparable behavior, albeit to a lesser extent.

**Visual Interpretations.** Fig. 22, 23, and 24 present examples of heatmaps depicting logit scores for the predicted class. We observe that $Z_{SY}$ appears noisier and less focused compared to $Z_Y$ and $Z_S$. This could be attributed to the challenges in visually representing associations between concepts. Nonetheless, the critical regions often encompass both the background and the object class. Consistent with textual representations, $Z_Y$ and $Z_S$ exhibit high concentration around the object and surrounding background. Additionally, the $Z_Y$ states show reduced noise compared to the overall image representation. Between models, the prediction heatmaps are noisier in ViT-H, which may be due to the larger set of states examined; see Tab. 14.

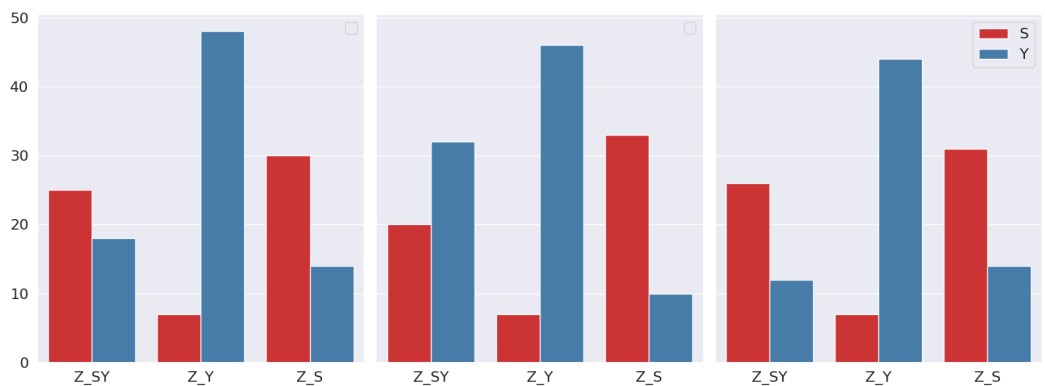

Figure 21: Normalized SHAP values towards text features belonging to $Y$ and $S$. Left: ViT-B/16, Middle: ViT-L/14, Right: ViT-H/14 **Dataset: Waterbirds**

Table 11: Top text features for $Z_{SY}, Z_Y, Z_S$. Dataset: Waterbirds. Model: **ViT-B/16 and ViT-L/14**

| | ViT-B/16 | | | ViT-L/14 | |
| --- | --- | --- | --- | --- | --- |
| $Z_{SY}$ | $Z_Y$ | $Z_S$ | $Z_{SY}$ | $Z_Y$ | $Z_S$ |
| forest | tern | forest | beak | tern | forest |
| bamboo | gull | bamboo | warbler | warbler | bamboo |
| beach | warbler | beach | gull | gull | beach |
| pond | beak | lake | tern | sparrow | lake |
| lake | sparrow | pond | sparrow | wren | pond |
| warbler | wren | river | feathers | cormorant | ocean |
| water | bill | ocean | wings | beak | trees |
| river | feathers | costal | bamboo | black | river |
| beak | woodpecker | water | lake | kingfisher | sunset |
| wings | duck | moss-covered | forest | woodpecker | shoreline |

Table 12: Top text features for $Z_{SY}, Z_Y, Z_S$. Dataset: Waterbirds. Model: **ViT-H/14.**

| | ViT-H/14 | |
|---|---|---|
| $Z_{SY}$ | $Z_Y$ | $Z_S$ |
| forest | tern | bamboo |
| beach | gull | forest |
| pond | warbler | beach |
| lake | beak | ocean |
| water | sparrow | lake |
| river | blue | pond |
| sunset | wren | river |
| grassy | black | sunset |
| trees | woodpecker | trees |
| shoreline | kingfisher | shoreline |

Table 13: Correlation between TextSpan (Gandelsman et al., 2023) and located states. For each state, the given text statements represents the top 5 textual descriptions that accounts for the variance across the Imagenet validation set. L10H10 denotes attention head at layer 10 and head 10. **Dataset: Waterbirds**

| Model | Top Localized States | | |
|---|---|---|---|
| | $Z_{SY}$: **L10H10** | $Z_Y$: **L11H5** | $Z_S$: **L11H6** |
| **ViT-B/16** | Tranquil boating on a lake | Photo of a reptile | Photo taken in Namib Desert |
| | Peaceful rural farmland | Image with a seagull | Photo taken in the Alaskan mountains |
| | Serene garden pond | An image with dogs | A photo of Monaco |
| | Secluded beach cove | Snapshot of a marsupial | Image taken in the Florida Everglades |
| | Picture taken in the Italian pasta kitchens | A thistle | contemplative coastal view |
| | $Z_{SY}$: **L23H14** | $Z_Y$: **L23H2** | $Z_S$: **L22H2** |
| **ViT-L/14** | An image with dogs | Image showing prairie grouse | Urban park greenery |
| | Majestic soaring birds | Image with a penguin | cozy home interior |
| | Graceful swimming fish | A magnolia | Urban subway station |
| | An image with bikes | An image with dogs | Energetic street scene |
| | Picture with boats | An image with cats | Tranquil boating on a lake |
| | $Z_{SY}$: **L30H11** | $Z_Y$: **L31H2** | $Z_S$: **L28H11** |
| **ViT-H/14** | calming riverbank scene | detailed reptile close-up | A bamboo |
| | Pristine snowy landscape | Image with polka dot patterns | A picture of a baby |
| | peaceful meadow landscape | A spiky texture | Photo taken in the Italian vineyards |
| | Sandy beach shores | Artwork featuring zebra stripe motifs | Blurred boundaries |
| | Gritty urban street scene | Image with a sheep | delicate soap bubble display |

Table 14: Individual state representations within $Z_Y$ for **ViT-B/16 and ViT-L/14.** The overall score for $Z_Y$ is shown beside each model while the SHAP score for individual states within $Z_Y$ is shown beside each head. The scores refer to **species/features**. The overall score is measured by first aggregating the state activations corresponding to $Z_Y$ before implementing SHAP, different from individual state activations. **Dataset: Waterbirds**

| ViT-B/16: 44/15 | | ViT-L/14: 44/12 |
|---|---|---|
| L11H5: 44/14 | L10H2: 21/19 | L23H2: 44/12 |
| tern | beak | tern |
| gull | tern | warbler |
| warbler | warbler | gull |
| beak | feathers | sparrow |
| sparrow | wings | wren |
| wren | woodpecker | bamboo |
| feathers | bamboo | cormorant |
| woodpecker | white | beak |
| bill | breasted | black |
| duck | jay | kingfisher |

Table 15: Individual state representations within $Z_Y$ for **ViT-H/14.**

| ViT-H/14: 38/19 | | | |
|---|---|---|---|
| L30H8: 39/16 | L31H2: 39/12 | L31H1: 36/17 | L31H13: 20/25 |
| tern | tern | tern | yellow |
| gull | gull | gull | red |
| warbler | warbler | warbler | black |
| beak | bamboo | black | tern |
| sparrow | sparrow | sparrow | blue |
| wren | beak | white | white |
| feathers | wren | wren | brown |
| woodpecker | woodpecker | yellow | green |
| bill | feathers | blue | orange |
| duck | kingfisher | brown | sunset |

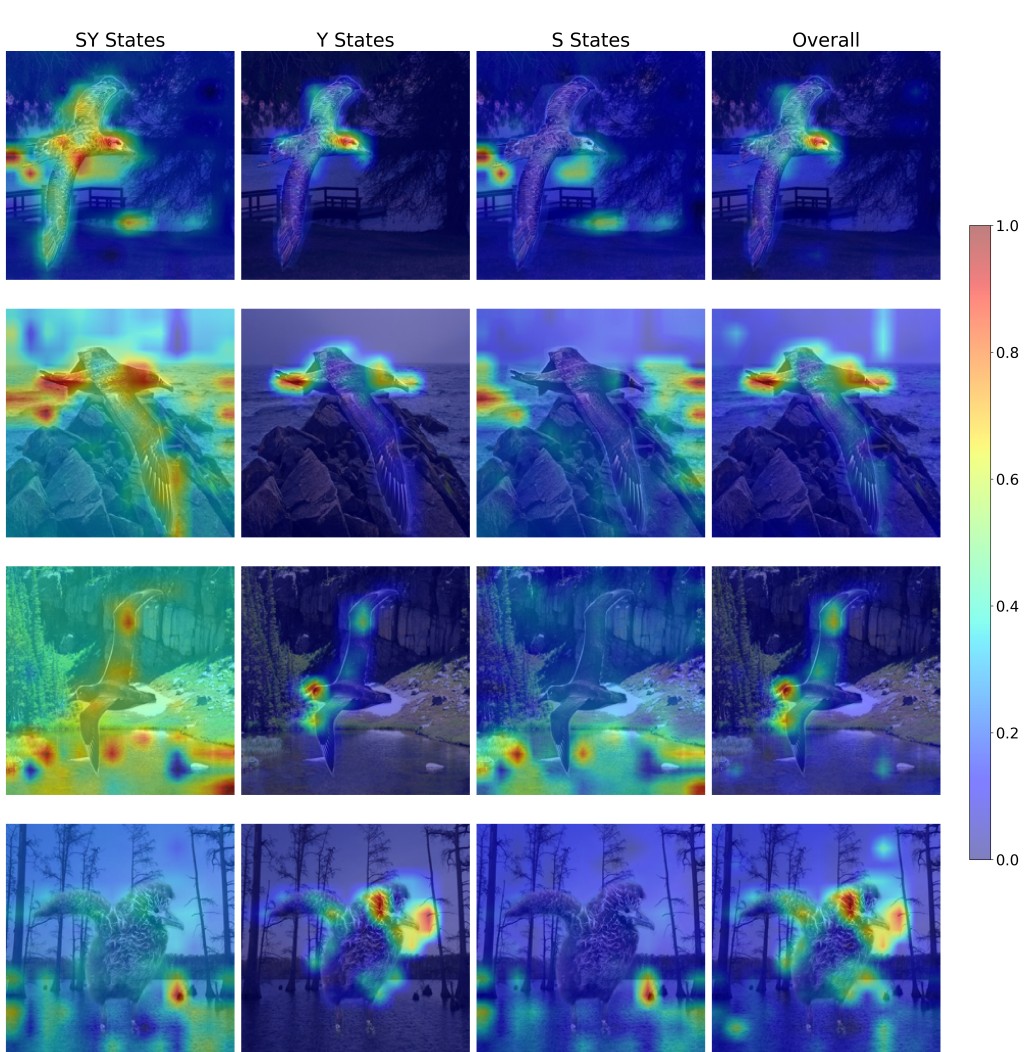

Figure 22: Image visualization: Localized representations of $Z_{SY}, Z_Y, Z_S$ and overall image. **Model: ViT-B/16. Dataset: Waterbirds**

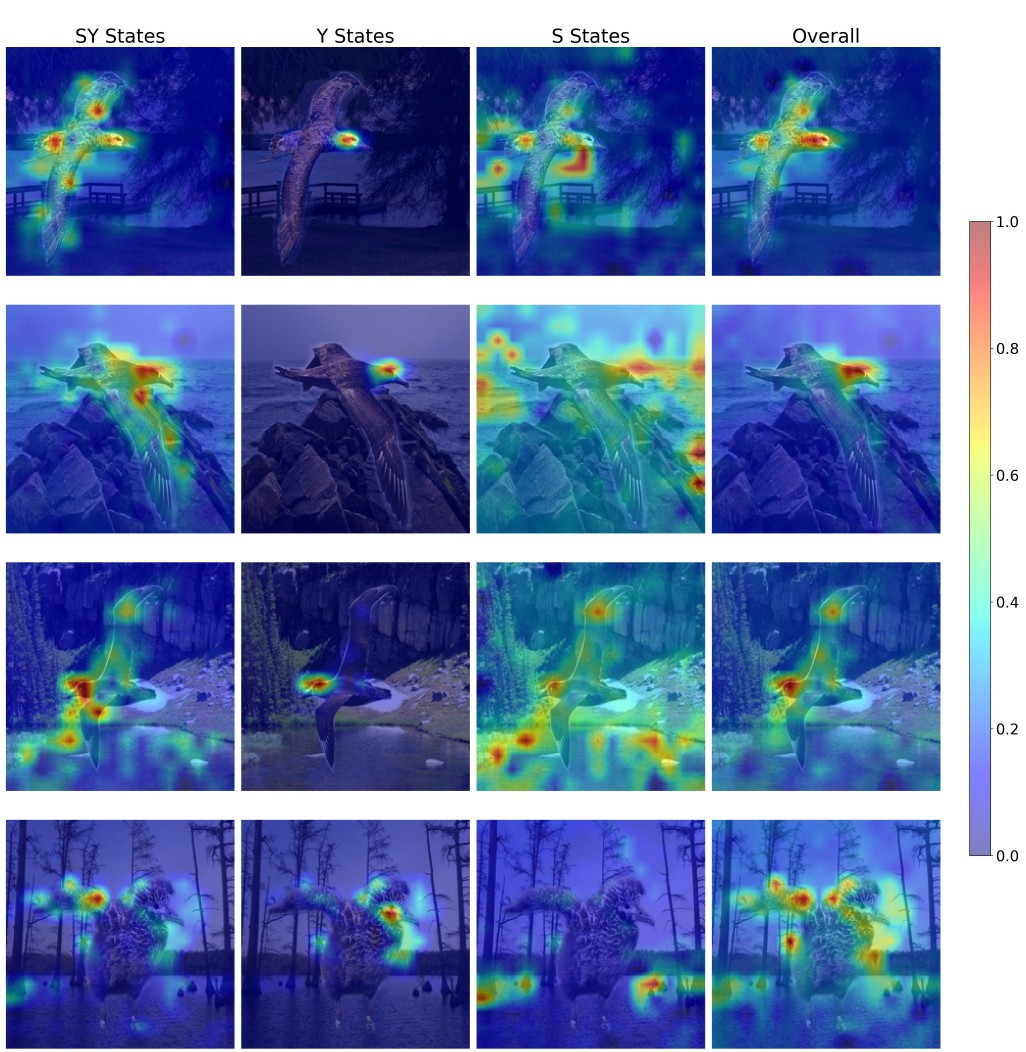

Figure 23: Image visualization: Localized representations of $Z_{SY}, Z_Y, Z_S$ and overall image. **Model: ViT-L/14. Dataset: Waterbirds**

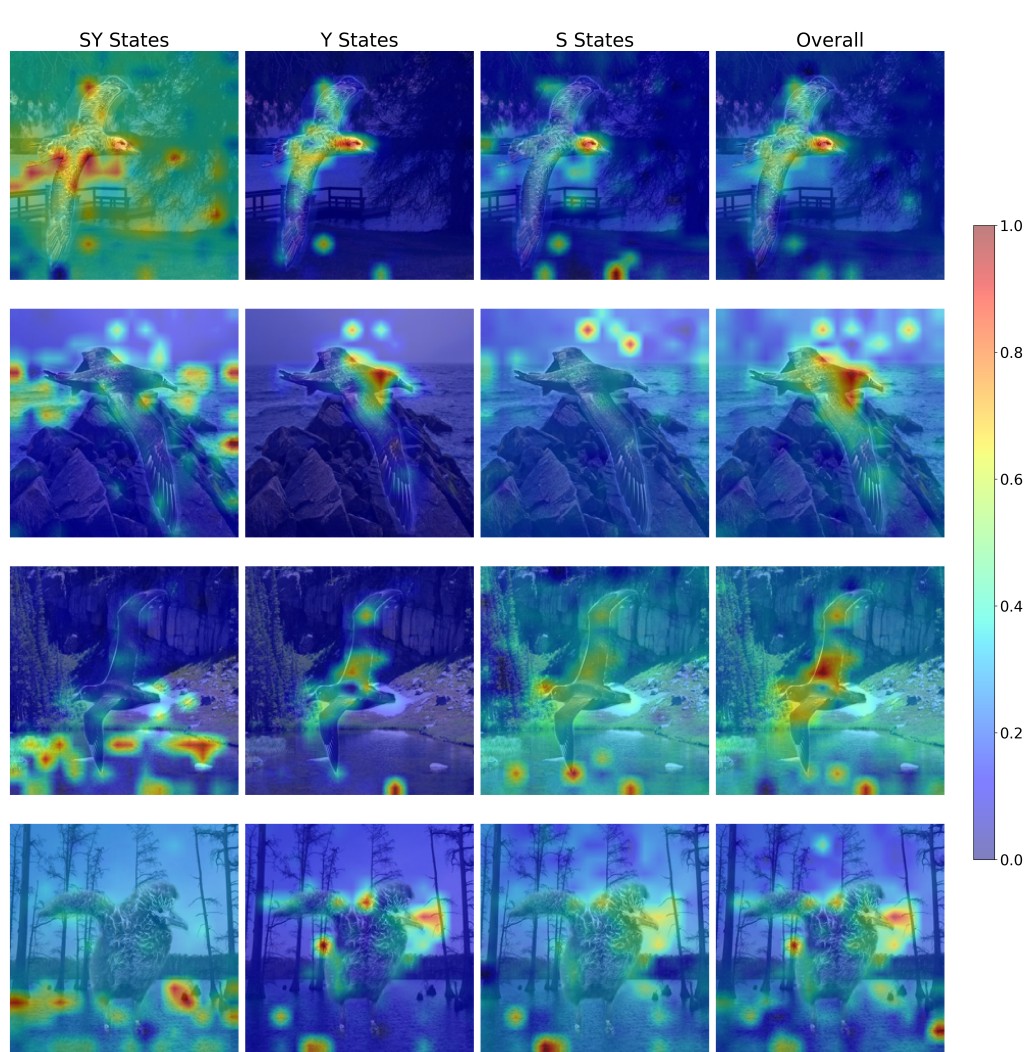

Figure 24: Image visualization: Localized representations of $Z_{SY}, Z_Y, Z_S$ and overall image. **Model: ViT-H/14. Dataset: Waterbirds**

## D.2 GENDERBIAS

Fig. 25 presents the SHAP scores, revealing findings consistent with those from the Waterbirds dataset, where score ratios align with the corresponding attributes. However, $Z_S$ exhibits a lower ratio, $\frac{S}{Y}$, compared to Waterbirds. This may be due to $Y$, representing gender, occupying only a single token in the caption, unlike attributes such as occupation or feature category. Despite gender descriptions being present in all captions, Tab. 16 shows that no gender-related features appear in $Z_Y$ across any of the models.

**Correlation with TextSpan.** We observe a positive correlation with TextSpan in the GenderBias dataset, where the descriptions of $Z_S$ align with gender-related attributes or references to people. In contrast, the descriptions in $Z_Y$ are associated with occupational objects or visual depictions of work settings, such as "*bioreactor*" and "*dance pose*", which likely contribute to classifying the occupation depicted in the image.

**Individual state representation.** Most tokens in Tab. 18, 19 and 20 relates to descriptions of occupational equipment. For states with a higher occupation-to-feature ratio, we see higher occurrences of occupational terms. This aligns with the expectation that surrounding objects in an image are critical for classifying a person's occupation. For instance, a stethoscope serves as a key medical device to differentiate between a doctor and a nurse, and its inclusion becomes even more significant in mitigating bias when classifying a male nurse.

**Visual Interpretations.** We observe more plausible heatmaps on ViT-B compared to the larger models. Most examples align with human intuition when classifying the occupation depicted in an image, focusing on elements such as "*helmet, workplan: civil engineer*", "*desk with computer: receptionist*", or "*computer: web developer*". In contrast, larger models exhibit noisier distributions that may not align with human reasoning. This observation highlights a potential limitation in identifying states that do not necessarily encode the hypothesized attribute effectively.

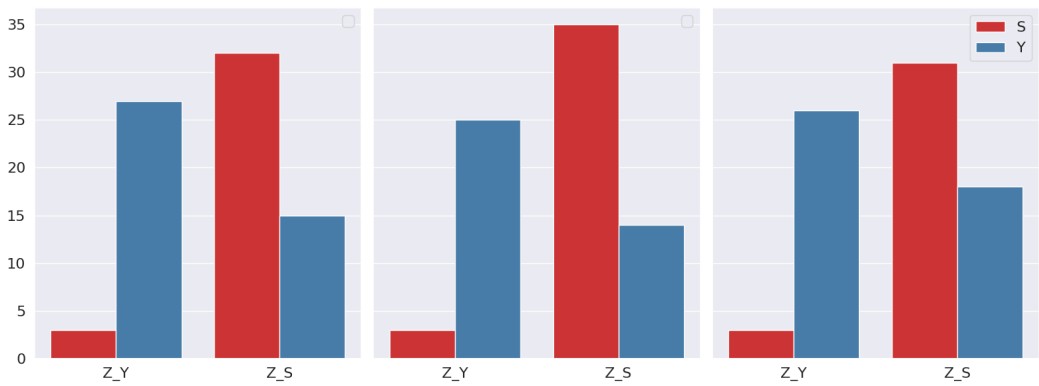

Figure 25: Normalized SHAP values towards text features belonging to $Y$ and $S$. Left: ViT-B/16, Middle: ViT-L/14, Right: ViT-H/14 **Dataset: Genderbias**

Table 16: Top text features for $Z_{SY}, Z_Y, Z_S$. Dataset: Genderbias.

| ViT-B/16 | | ViT-L/14 | | ViT-H/14 | |
|---|---|---|---|---|---|
| $Z_Y$ | $Z_S$ | $Z_Y$ | $Z_S$ | $Z_Y$ | $Z_S$ |
| office | male | office | male | office | male |
| desk | female | desk | female | desk | female |
| laptop | manager | correctional | worker | computer | supervisor |
| police | her | construction | his | documents | his |
| physician | his | detectives | technician | headset | worker |
| monitors | cheerful | laundry | her | physician | her |
| officers | mechanic | factory | mechanic | construction | manager |
| laundry | smiling | workshop | manager | medical | mechanic |
| computer | worker | hospital | administrator | correctional | suit |
| medical | she | industrial | supervisor | laundry | uniform |

Table 17: Correlation between TextSpan (Gandelsman et al., 2023) and located states. For each state, the given text statements represents the top 5 textual descriptions that accounts for the variance across the Imagenet validation set. L10H10 denotes attention head at layer 10 and head 10. **Dataset: Genderbias**

| Model | Top Localized States | |
|---|---|---|
| **ViT-B/16** | $Z_S$: **L11H4** | $Z_Y$: **L11H8** |
| | Image with a five people | A laptop |
| | Quirky street performer | A rug |
| | An image with dogs | A shelf |
| | Image with three people | A bookmark |
| | A photo of a woman | A bag |
| **ViT-L/14** | $Z_S$: **L23H4** | $Z_Y$: **L23H1** |
| | Playful siblings | Photograph taken in a retro diner |
| | A photo of a young person | Intense athlete |
| | Image with three people | Detailed illustration of a futuristic bioreactor |
| | A photo of a woman | Image with holographic retro gaming aesthetics |
| | A photo of a man | Antique historical artifact |
| **ViT-H/14** | $Z_S$: **L31H7** | $Z_Y$: **L31H6** |
| | A photo of a woman | Evocative dance pose |
| | A photo of a man | Picture with cars |
| | Energetic children | A photo of food |
| | An image of a couple | Graceful swimming fish |
| | Warm home interior | thrilling sports action |

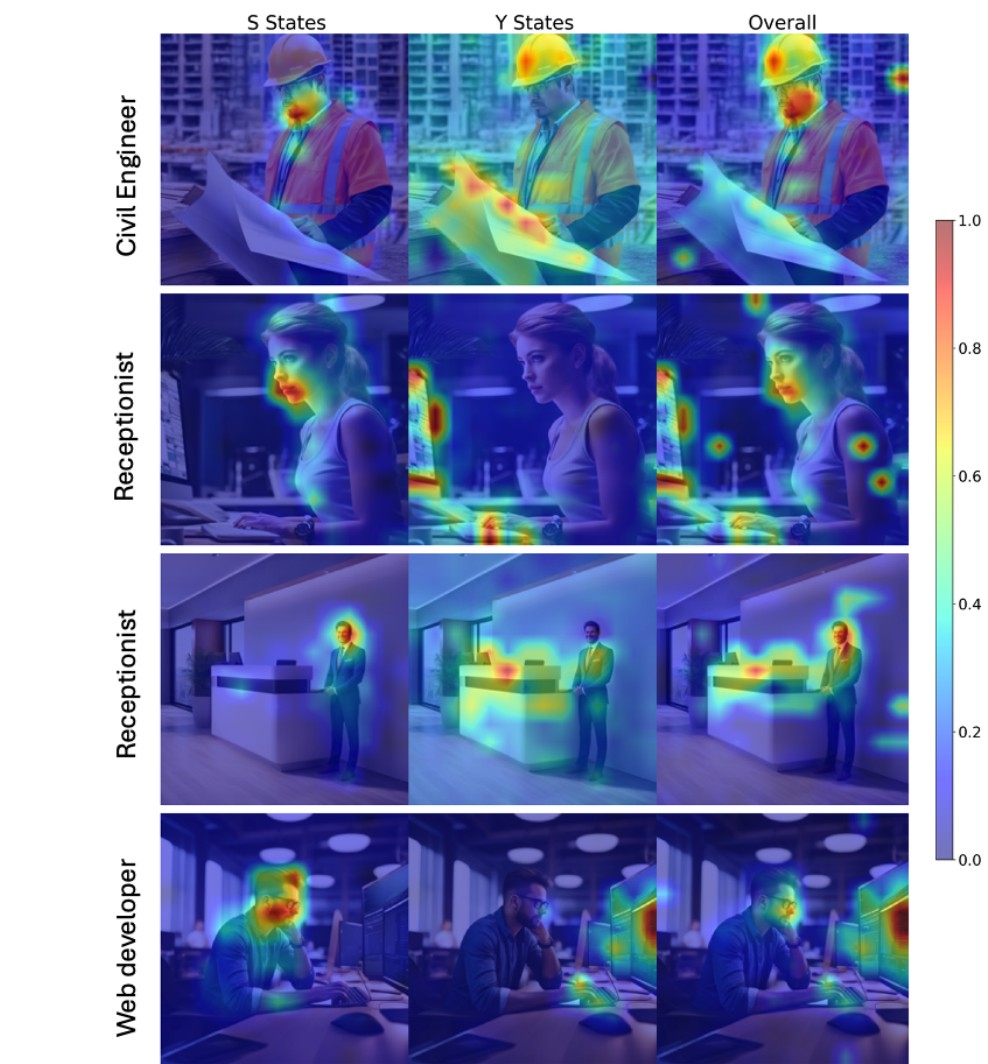

Figure 26: Image visualization: Localized representations of $Z_Y, Z_S$ and overall image. **Model: ViT-B/16. Dataset: Genderbias**

Table 18: Individual state representations within $Z_Y$ for **ViT-B/16**. Scores refer to **occupation/features**. **Dataset: Genderbias**

| ViT-B/16: 23/15 | | |
|---|---|---|
| L11H3: 25/13 | L11H5: 20/13 | L11H8: 16/16 |
| office | manager | office |
| physician | headset | desk |
| police | firstline | laptop |
| desk | office | computer |
| officers | stethoscope | documents |
| laundry | refractory | monitors |
| administrator | stacks | coat |
| medical | computer | screens |
| workers | supervisor | firstline |
| hospital | tie | laundry |

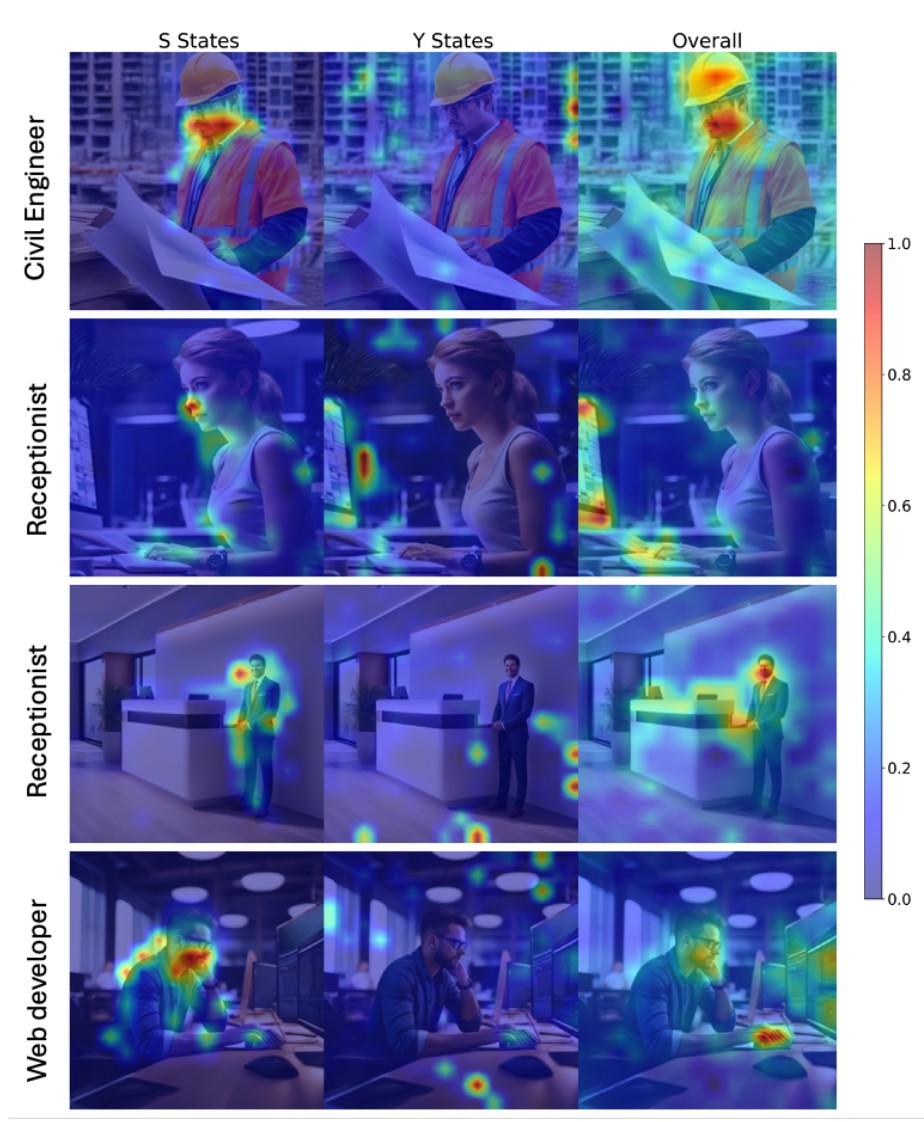

Figure 27: Image visualization: Localized representations of $Z_Y, Z_S$ and overall image. **Model: ViT-L/14. Dataset: Genderbias**

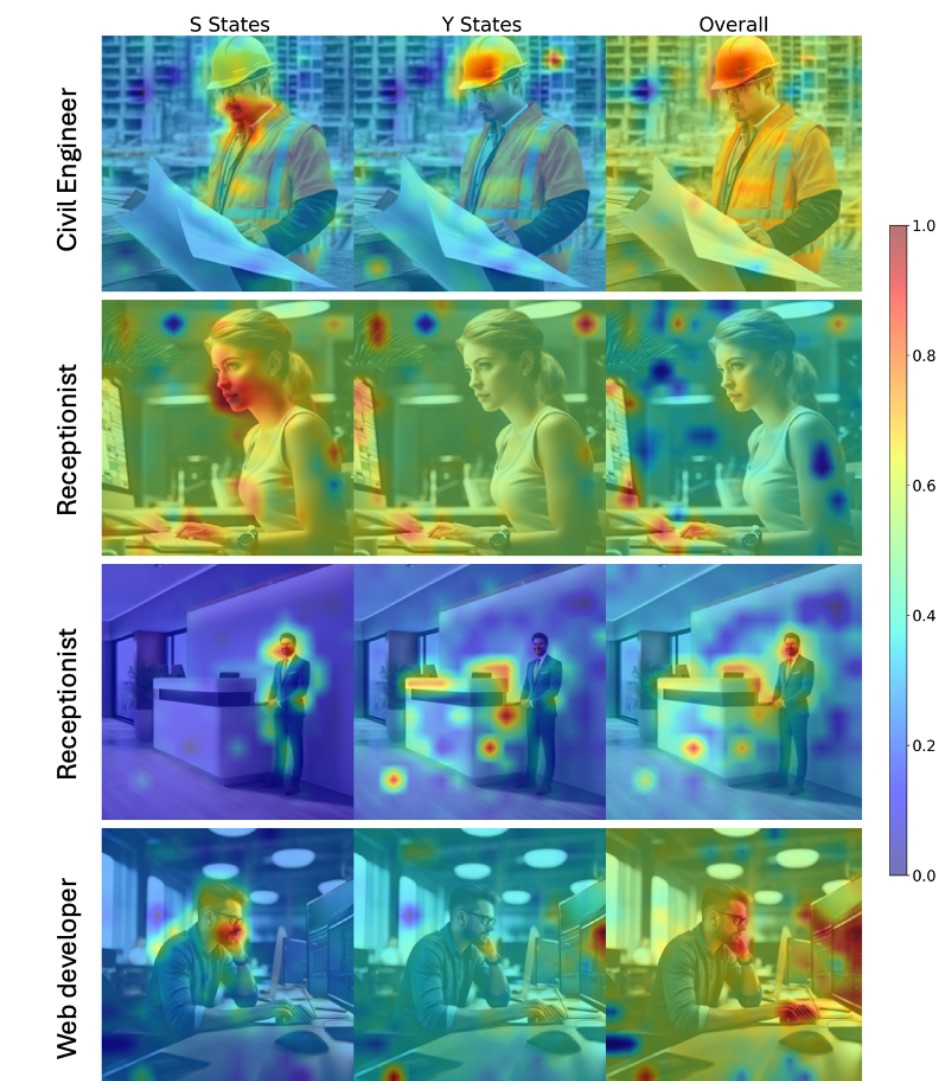

Figure 28: Image visualization: Localized representations of $Z_Y, Z_S$ and overall image. **Model: ViT-H/14. Dataset: Genderbias**

Table 19: Individual state representations within $Z_Y$ for **ViT-L/14**. Scores refer to **occupation/features**. **Dataset: Genderbias**

| ViT-L/14: 23/13 | | | |
|---|---|---|---|
| L23H1: 26/13 | L22H2: 19/11 | L23H3: 18/11 | L23H12: 21/12 |
| office | office | office | firstline |
| desk | desk | desk | data |
| laptop | laptop | factory | office |
| refractory | laundry | facility | stethoscope |
| correctional | construction | correctional | computer |
| detectives | lab | factory | manager |
| physician | room | workshop | documents |
| headset | industrial | lab | laptop |
| administrator | female | construction | refractory |
| laundry | workshop | supervisor | correctional |

Table 20: Individual state representations within $Z_Y$ for **ViT-H/14**. **Dataset: Genderbias**

| ViT-H/14: 23/15 | | | |
|---|---|---|---|
| L31H6: 21/14 | L30H7: 22/14 | L30H13: 20/13 | L30H12: 21/10 |
| office | computer | office | manager |
| desk | suit | documents | computer |
| computer | correctional | suit | male |
| headset | construction | refractory | female |
| medical | laptop | paperwork | correctional |
| suit | documents | desk | exuding |
| officers | lab | correctional | worker |
| laundry | laundry | firstline | physician |
| construction | desk | telecommunicator | confident |
| police | electronic | office | office |

# E    COMPUTATIONAL REQUIREMENTS

All experiments can be ran with a single Nvidia A100 80GB GPU. Since the only training involved is on a 2-layer classifier, our work does not involve heavy computational usage.

# F    SOCIETAL IMPACT

While our work focuses on mitigating bias in vision-language models, we acknowledge that the underlying methods could, in principle, be reversed to amplify spurious correlations—such as reinforcing implicit gender biases. However, we do not consider this a significant risk in the context of our work, as the methodology requires explicit identification and manipulation of known components and would unlikely be possible on proprietary models. Rather than enabling harm, we believe our approach advances the growing literature on bias mitigation and promotes transparency by shedding light on how biases arise from large-scale pretraining or fine-tuning on imbalanced datasets.

