# OpenReview forum: "Debiasing CLIP: Interpreting and Correcting Bias in Attention Heads"
_ICLR.cc/2026/Conference — Submitted to ICLR 2026_

### Official Review · Reviewer_dz5L · 2025-10-17

**Soundness:** 3
**Presentation:** 2
**Contribution:** 3
**Rating:** 6
**Confidence:** 4

**Summary:**

This paper interprets and mitigates biases in CLIP from a mechanistic perspective by focusing on biased attention heads. The authors propose a novel framework, LTC, to measure the contribution of each attention head across layers towards the final prediction, and further isolate spurious and target contributions. With such insights, the authors then propose to debias by suppressing spurious attention states while enhancing target states. Experiments across various CLIP backbones and different bias types show the effectiveness and generalizability of LTC. The text interpretation and image interpretation results also further validate the interpretability of LTC.

**Strengths:**

1. Unlike existing methods which study the biases in the overall image or text embeddings, the proposed idea of using mechanistic interpretability to probe biases in ViT attention heads is novel and provides a new perspective for bias interpretation in CLIP
2. In terms of debiasing, LTC outperforms existing methods (e.g., Ortho-Cali and Roboshot) across various benchmarks , further closing the gap between subgroups.
3. The interpretation results, including text interpretation based on SHAP values and image interpretation based on pixels, provide coherent explanations of the SY, Y and S states. These findings further validate the interpretability of LTC.
4. LTC supports both non-parameter-tuning and parameter-tuning setups, suitable for a wide range of debiasing needs
5. Extensive debiasing, interpretability and ablation experiments across various CLIP backbones offer strong support of the effectiveness and interpretability of LTC on CLIP models

**Weaknesses:**

1. The proposed LTC framework is only for interpreting and debiasing attention head activations of the image representations. The biases in text representations, on the other hand, are not studied and addressed, thereby limiting the framework's ability to comprehensively address potential multi-modal biases in CLIP. Can the proposed LTC be further extended to text encoder of CLIP, which is also transformer-based and has attention heads?
2. Since LTC assumes a transformer-based architecture for image encoders,  such as ViT, it cannot be applied on non-transformer-based CLIP variants, such as CLIP-ResNet, thus limiting the applications of LTC.
3. The modeling of spurious and target state contributions in Sect. 4.2 is an interesting approach to decouple target and spurious states, but further clarification is required for the contrastive solution to V_S described in Eq. 10: specifically, how the inequalities in 220 lead to the monotone relations in line 223 after "normalizing each sample's contributions to unit mass" is not clear; additional explanations/derivations on how line 223 inequalities support the formulation of V_S in Eq. 10 will also be helpful for readers' understanding of Sect. 4.2.
4. A minor suggestion is that more results and analyses on attention states interpretation can be included in the main paper, especially on the "spurious association" section: additional in-depth analysis on Z_SY, "the knowledge of associating S with Y", will provide more insights on the complex nature of biases in CLIP. Can Z_SY be also suppressed in the same way shown in Alg. 1 for additional bias mitigation?

**Questions:**

Please refer to the weakness part.

---

> ### Author Response · Authors · 2025-11-17
> **Clarifications on reviewer's concerns (Part 1)**
>
> Dear Reviewer dz5L,
>
> Thank you for your time in providing constructive feedbacks and comments. We are glad that you found our work to be novel and has provided a new perspective on debiasing! We will address your concerns below.
>
> **1. Extension towards text representations**
>
> Yes that is correct! LTC is definitely extensible towards text representations, since the text decoder is also a transformer architecture and Eq2-4 would be applicable. However, since the main focus of our work is to demonstrate the effectiveness of the two-stage pipeline, localizing decomposed states before performing corrective measures (debiasing or knowledge injection). Thus we leave experiments on jointly performing LTC to both image and text representations to future work, we believe there are definitely more positive results to expect!
>
> **2. Limited applicability beyond transformer models**
>
> We agree that this is definitely a limitation in our work and we thank the reviewer for raising it, we have added this to our limitation section in the revised manuscript (highlighted in red). However, we do not view this as a significant issue since ViT is widely used over ResNet and unlike LLMs, CLIP ViTs  are much more lightweight, containing less than 1B parameter for the biggest model (ViT-H).

---

> > ### Comment · Reviewer_dz5L · 2025-11-27
> > **Response to authors**
> >
> > I have read the authors' rebuttal and also see other reviewers' concerns.
> > It seems that some important related work is missing.
> >
> > In the meantime, I saw some merits in this work, and therefore keep my borderline accept score.
> >
> > However, I'm afraid I will not champion this work during our discussion with reviewers.
> >
> > In addition, please fold the responses into fewer blocks so that there will be fewer notifications.

---

> ### Author Response · Authors · 2025-11-17
> **Clarifications on reviewer's concerns (Part 2)**
>
> **3. Clarifications on spurious and target contributions**
>
> That is a great question! Firstly, the derivations of the contributions in L220 is conducted by primarily focusing only on samples within $G_N$ (samples whose spurious attribute would cause a misclassification, such as waterbirds on land background - causing landbird to be predicted due to the background). The high-level rationale behind Eq.10 is that we believe it is more than likely that the spurious heads (encode the spurious attribute; land background) have a higher contribution than the target heads (encode the class itself; waterbirds) if **the model failed to predict the correct class; waterbirds** and vice versa target heads having a higher contribution when predicted correctly. Thus implementing this equation over a dataset should sufficiently filter out the $S$ and $Y$ head.
>
> Now with respect to the inequalities in L220, it follows from Eq.8 where we decompose the prediction onto the $S$ and $Y$ concept subspace, with Eq.8 being $V = V_Y - V_S$ for $G_N$ (as explained above). Following this, in $V_{NW}$ (samples predicted wrongly in $G_N$) the RHS of Eq.8 is negative if we assume $y = y^*$ in Eq6, and similarly positive for $V_{NC}$ (predicted correctly in $G_N$). Comparing the two inequalities would led us to L220, which can be interpreted for $V_{S|G_{NW}} > V_{Y|G_{NW}}$ as “spurious heads activate more than target heads, thus model predicted class label that is spuriously correlated with the spurious attribute” and vice versa for $V_{Y|G_{NC}} > V_{S|G_{NC}}$. After normalizing Eq 8 to summed up to $1$, we know for certain that $V_S$ in $G_{NW}$ would definitely have a higher normalized direct effect than $G_{NC}$, which is intuitive since this is how the model predicts wrongly. Finally, since in Eq.10, we filter out negative effects, by contrasting the two, we would effectively cancel out $V_Y$ for $V_{NW} - V_{NC}$. Likewise, flipping the two terms cancel out $V_S$.
>
> We hope that this has sufficiently clarified the reviewer’s concern regarding the derivations, else we will be happy to provide further answers!

---

> ### Author Response · Authors · 2025-11-17
> **Clarifications on reviewer's concerns (Part 3)**
>
> **4. Spurious Association states for debiasing**
>
> Thank you for the excellent question! To make it clear, we did use $Z_SY$ for the debiasing, with the caveat that we are able to separate heads that encode $SY$ from heads that  encode $S$; yes in waterbirds and no for genderbias. L423-440 shows the following:
>
> We discovered that the identified $Z_S$ using Eq10 was actually $Z_{SY}$. This discovery is proven by being able to find a different set of heads that actually encode $Z_S$ which is then proven by causal experiments; ablating $Z_{SY}$ does not affect classification of $S$. Based on how we structure our contrastive learning, this is intuitive. The high-level idea of Eq.10 is to discover reasons (concepts encoded) that causes the model to wrongly classify the target. For  $Z_{SY}$, the concept would be: “associating the target class with the spurious class” and “predicting background class” for $Z_S$. Thus, what we discovered was that we were actually ablating the attention heads that encode the association between the two concepts rather than an individual concept.
>
> This is actually a very interesting discovery, in part thanks to the proposed contrastive solution. Most existing works sought to ablate the spurious concept entirely, which is a cheap way of solving the problem. However, the root cause is not due to the model being able to identify $S$ but rather, learning the shortcuts of associating $S$ with $Y$. A drawback of taking this shortcut is that it affects the classification on $Z_S$ - which most of these works ignore by assuming that the model will not have to classify $S$. However, in this work, we made no such assumption.
>
> Once again, thank you for your time and we are extremely grateful for the intriguing questions asked. We are pleased with the interesting questions asked and hope we have provided sufficient clarifications. We would greatly appreciate it if the reviewer can increase the rating accordingly if the concerns have been appropriately addressed.

---

> ### Author Response · Authors · 2025-11-27
> **Kind reminders**
>
> Dear Reviewer dz5L,
>
> We sincerely thank the reviewer once again for their careful reading and constructive feedback, which have helped us significantly improve the quality and clarity of this work. We have addressed all raised concerns in detail in the rebuttal and corresponding revisions. We kindly invite the reviewer to reassess the manuscript in light of these changes, and if the responses have satisfactorily resolved the concerns, we would greatly appreciate reconsideration of the scores accordingly.

---

### Official Review · Reviewer_2p5V · 2025-10-28

**Soundness:** 4
**Presentation:** 4
**Contribution:** 2
**Rating:** 4
**Confidence:** 4

**Summary:**

This paper tackles the problem of bias in multimodal models like CLIP, where they learn spurious associations, such as linking a "waterbird" to "water". The authors propose Locate-Then-Correct (LTC), a framework that identifies the specific attention heads in the model responsible for the bias and those relevant to the task. By ablating the spurious heads and enhancing the task-relevant heads with orthogonal projection, LTC achieves over a 50% gain in worst-group accuracy on bias benchmarks compared to other post-hoc methods.

**Strengths:**

1. The paper is very well-written, well-presented, and easy to follow.
2. The methodology is technically solid, with clear and correct mathematical notations.
3. Illustrations and tables are clear and effectively support the paper's claims.

**Weaknesses:**

Despite the paper's strengths, the primary weakness lies in the discussion and comparison to related work. While the application of Causal Mediation Analysis is interesting, the core "locate-and-correct" idea is not unique. The authors, perhaps due to the timing of publication, seem to have missed several recent and highly relevant papers that explore similar ideas and show strong performance in debiasing.

The authors should discuss these papers and provide a clear rationale for why the proposed method is novel or advantageous in comparison. At a minimum, performance should be benchmarked against them.

Missing related works include:

- "Debiasing attention mechanism in transformer without demographics", Lu et. al., ICLR 2024
- "A Unified Debiasing Approach for Vision-Language Models across Modalities and Tasks", Jung et. al., NeurIPS 2024
- "NeuronTune: Towards Self-Guided Spurious Bias Mitigation", Zheng et. al., ICML 2025
- "EvA: Erasing Spurious Correlations with Activations", He et. al., ICLR 2025

On the other hand, the idea using direct effect for ViT for debiasing has been discussed in this new paper:
- "Model Editing for Vision Transformers", Huang et. al., NeurIPS 2025 (this paper might be accepted after ICLR submission though)


As the basic idea looks similar to me, authors should discuss these papers, and give audience to proper rationale why the proposed method is better than the papers above. If you don't find proper rationale, at least the performance should be better than them.



## Minor Point
The core of the proposed method is Direct Effect (from Causal Mediation Analysis). My concern here is twofold:
1. Technically, Direct Effect is not synonymous with an "importance score."
2. More fundamentally, I question the theoretical justification for applying Causal Mediation Analysis to a deep neural network. While I recognize that many empirical papers have used this approach, its theoretical validity is questionable. Neural networks are high-dimensional, non-linear systems that do not inherently represent a formal causal graph, which is a foundational assumption for CMA.

While a full theoretical defense is likely out of this paper's scope, the authors should be aware of this methodological limitation. As an out-of-scope discussion, the concern related to CMA doesn't affect my rating.

**Questions:**

I have one high-level question: If the authors chose to use causal effects as the basis for their method, why limit it to Transformer-based models? Causal effects can be (and have been) applied to CNNs as well (e.g., "Mediation CNN (Med-CNN) Model for High-Dimensional Mediation Data").

What was the specific technical justification for narrowing this approach to attention-based methods only, rather than applying it more broadly?

---

> ### Author Response · Authors · 2025-11-17
> **Clarifications on reviewer's concerns (Part 1)**
>
> Dear reviewer 2p5V,
>
> Thank you for your time in providing constructive feedbacks and comments. We are glad that you found our work to be well-written and presented with solid mathematical notations. We will address your concerns below.
>
> **1. Related works**
>
> We thank the reviewer for introducing the relevant works. However, while these works do perform some form of localization, they have fundamental differences from our LTC framework. We discuss the differences below.
>
> - “Debiasing Attention Mechanism in Transformer without Demographics.”: LTC identifies which specific ViT heads carry spurious vs. target signal (via causal mediation/contrast) and edits only those heads (mean-ablate S; inject Y) at test time. Whereas, this work modifies attention everywhere uniformly; **it’s not a locate-then-edit, and it doesn’t provide per-head causal attributions. Moreover, this work requires training, whereas LTC is applicable on both inference and training settings.**
> - “A Unified Debiasing Approach for VLMs (SFID).”: The main difference from our work is that **this is a global representation level type of debiasing**, whereas in LTC, we take a mechanistic approach, localizing the debiasing to specific attention heads and with clear reasons why we do so (heads are interpreted to encode a certain concept, $S$, $Y$ or $SY$ in Chapter 6)
> - “NeuronTune: Towards Self-Guided Spurious Bias Mitigation.”: This work does not target attention heads but rather only specific neurons in DNNs for debiasing. **Besides localizing components for debiasing, LTC also performs an additional knowledge injection to impair target-specific concept knowledge.** Moreover, this method is a training-only method, and cannot be applied on inference time only. Most importantly, there is no interpretability component here, e.g. the work does not discuss what these neurons represent, just that it improves spurious correlation.
> - "EvA: Erasing Spurious Correlations with Activations.”: This work learns class-specific spurious indicators inside pretrained networks and erases them as a debiasing step. Once again, the key difference lies with being on the **global activation-space rather than specific attention heads.**
> - “Model Editing for Vision Transformers (RefineViT)”: As the reviewer correctly pointed out, this work is indeed released after ICLR submission, thus we would not have been aware of it. However, a key difference is that our framework discover the spurious heads differently. **In their case, RefineViT took a rather naive approach by ablating each head independently and taking top $K$. In our framework , we take a contrastive approach**. This is based on the simple assumption that “spurious heads” would be more active on samples with opposite spurious-target attribute (‘waterbirds against land”) if **model predicts wrongly** and “target heads” if **predicted correctly**. Empirically (See below), we find that our method, **LTC outperforms RefineViT by a huge margin**, proving that the attention heads identified is cleaner.
>
> In RefineViT, they train the projection matrix while we either do not train any parameters under non-parameter-tuning setting or we train the classifier during parameter-tuning setting. **Thus, there is no similar setting in which both of these works can be compared fairly**. However, the difference between non-parameter and parameter setting is mainly related to absolute accuracy score rather than the performance gap between groups. Thus we use performance **GAP (lower is better)** as a comparison since this would isolate the raw classification ability and instead focus on inter-group robustness; the main goal of debiasing.
>
> | Method | ViT-B/16        |ViT-L/14 | ViT-H/14 |
> |:--------------|------:|------:|------:|
> | RefineViT     | 19.7  | 13.4  | 11.9 |
> | LTC (non-train)    | 10.2 | 8.5 | 3.7 |
> | LTC (train)    | 4.8 | 3.0 | 3.7 |
>
> We also note that our **evaluation is broader and more principled than that of RefineViT**. We benchmarked LTC against both training-based and training-free baselines and, crucially, provided a causal interpretation of what each head encodes, establishing why our corrective interventions work. This interpretive grounding builds trust in the edits . We hope that this rationale is able to address the reviewer’s concern.

---

> ### Author Response · Authors · 2025-11-17
> **Clarifications on reviewer's concerns (Part 2)**
>
> **2. Direct effect and Causal Mediation Analysis**
>
> Firstly, we like to note that we use the term “importance score” to tie the importance of a component with respect to the score associated with it. This is to provide means to select specific heads for either debiasing or improvement (knowledge injection). We hope this clarifies any confusion with the use of the term.
>
> Secondly, we do agree with the reviewer’s point on CMA that there are indeed some underlying assumption required a-priori when applied to DNNs. However, it is important to note that the direct effects derived are not strictly **causal mediation**, though we borrowed the term “direct effect” from the literature. Referencing Eq2-4, the direct effects **are simply derived via linear decomposition (possible due to residual connections in transformers)**. This is mathematically proven and sound. In other words, the model is **treated as a deterministic computation graph, not a probabilistic causal graph.** We hope that our explanation has addressed any confusions regarding the direct effect derivation.

---

> ### Author Response · Authors · 2025-11-17
> **Clarifications on reviewer's questions**
>
> **3. Why transformers instead of CNN**
>
> As the reviewer has mentioned, there are plenty of works that applied causal effects to CNN-type models (for example ResNet before ViT was popular). Thus, we do think much value can be applied in that area and instead focus on transformer where the mechanistic interpretability field has taken much interest in. We also want to clarify that it is not in our intention to focus on **attention** modules, but rather transformers due to the ease of deriving direct effect, made possible due to its architecture. The reason as to why attention heads are considered is because MLP layers were empirically shown to not contribute significantly in the image classification tasks (See L154).
>
> Once again, thank you for your time and we are extremely grateful for the intriguing questions asked. We are pleased with the interesting questions asked and hope we have provided sufficient clarifications. We would greatly appreciate it if the reviewer can increase the rating accordingly if the concerns have been appropriately addressed.

---

> ### Comment · Reviewer_2p5V · 2025-11-17
>
> Thanks for the prompt response. While I appreciate the clarification on the mechanistic differences, my concern regarding the comparative scope remains.
>
> In the paper (e.g., Line 44), the primary motivation is framed as "training-free" debiasing. While you do compare against RoboShot and Ortho-Cali, the justification for excluding other recent, high-performing training-free or inference-time approaches (such as SFID or the others mentioned) is unconvincing.
>
> The fact that LTC is 'mechanistic' or operates on attention heads is a methodological detail ('how' it works), not a distinct category of utility ('what' it achieves) that exempts it from comparison with global representation methods. If LTC is to be proposed as a superior tool for the task of debiasing, it must be benchmarked against the current state-of-the-art in that task, regardless of whether those methods operate on heads, neurons, or global embeddings.
>
> Without these comparisons, the paper's contribution is effectively limited to 'an interpretable CMA-based approach,' rather than a competitive 'training-free debiasing method.' To convince the audience that a user should choose LTC over methods like SFID or TargetBias, you must demonstrate superior performance or a distinct practical advantage beyond just the granularity of the edit.

---

> > ### Author Response · Authors · 2025-11-18
> > **Additional experiments on SFID**
> >
> > thank you for the swift response!
> >
> > From the suggested works, SFID and Eva are the only parameter-free methods, but since the code for Eva is not yet available, we will instead compare SFID as well as the previous RefineViT with LTC. Since the authors in SFID only studied genderbias, we compare with LTC on Genderbias-VL and RefineViT on waterbirds likewise for the same reason.
> >
> > For Waterbirds, we compare using performance gap between WG and AVG, lower is better.
> >
> > **Waterbirds (LTC vs RefineViT)**
> > | Method | ViT-B/16        |ViT-L/14 | ViT-H/14 |
> > |:--------------|------:|------:|------:|
> > | RefineViT     | 19.7  | 13.4  | 11.9 |
> > | LTC    | 10.2 | 8.5 | 3.7 |
> >
> > We compare two tasks, image classification and retrieval, we compare using the bias $B_T$ for classification and MaxSkew@10 $M_T$ for retrieval across the top 10 occupations, lower is better. B/L/H refers to Base/Large/Huge ViT backbones.
> >
> > **Genderbias-VL (LTC vs SFID)**
> > | Method | $B_O$   (B)     | $M_O$ (B)  | $B_O$   (L)     | $M_O$ (L)  | $B_O$   (H)     | $M_O$ (H)  |
> > |:--------------|------:|------:|------:|------:|------:|------:|
> > | SFID     | 26.4  | 29.9  | 35.5 | 30.9 | 41.2 | 26.6 |
> > | LTC | 10.0  | 17.2  | 18.0  | 20.4 | 27.4  | 24.5 |
> >
> > Across both background (LTC vs RefineViT) and gender (LTC vs SFID) bias benchmarks, we can see a clear improvement from LTC. Comparing against SFID, we can also see that LTC while being a much simpler framework, still produces remarkable results. Lastly, as discussed earlier, LTC provides an element of interpretability, whereas SFID is fully black-box.
> >
> > We hope these additional experiments are sufficient to convince the reviewer on the utility of our work and has sufficiently addressed the reviewer's remaining concerns, else we are happy to engage in further discussion, including the questions raised on CMA. We would deeply appreciate if the reviewer could raise the scores if these concerns have been adequately addressed.

---

> ### Author Response · Authors · 2025-11-27
> **Kind reminders**
>
> Dear Reviewer 2p5V,
>
> We sincerely thank the reviewer once again for their careful reading and constructive feedback, which have helped us significantly improve the quality and clarity of this work. We have addressed all raised concerns in detail in the rebuttal and corresponding revisions. We kindly invite the reviewer to reassess the manuscript in light of these changes, and if the responses have satisfactorily resolved the concerns, we would greatly appreciate reconsideration of the scores accordingly.

---

### Official Review · Reviewer_afvc · 2025-10-29

**Soundness:** 2
**Presentation:** 2
**Contribution:** 2
**Rating:** 4
**Confidence:** 3

**Summary:**

This paper introduces LOCATE-THEN-CORRECT (LTC), a novel, training-free framework designed to mitigate spurious correlations (such as background and gender biases) in Vision Transformer (ViT) based CLIP models. The authors evaluate LTC on benchmarks for background bias and gender bias. The results show that LTC outperforms other training-free baselines, achieving over a 50% gain in worst-group accuracy in some cases. The paper also provides extensive interpretability analyses (using SHAP and attention maps) to validate that the located heads indeed correspond to the hypothesized attributes.

**Strengths:**

1. Novel and Intuitive Location Method: The core contribution, the contrastive method for locating $P_S$ and $P_Y$ heads by comparing $G_{NC}$ and $G_{NW}$, is clever and well-motivated.
2. Targeted and Mechanistic Intervention: Unlike existing methods that apply a global correction to the final image or text representation, LTC performs intervention on specific attention heads. This mechanistic approach is more granular and effective.
3. Good Interpretability: The paper provides strong qualitative and quantitative evidence that the "located" heads are meaningful.

**Weaknesses:**

1. Clarity and Readability: The paper is difficult to read, with some parts of the exposition, particularly the dense methodology, being particularly difficult. Would benefit from clearer diagrams or worked-out toy examples.
2. Generalizability of Located Heads: The "Locate" step is inherently dataset-dependent, as it requires a set of samples (even if a validation set) to identify $P_S$ and $P_Y$. The paper shows one instance of generalization (reusing heads from GenderBias-VL for FairFace) and mentions reusing Waterbirds heads for CounterAnimal. However, the limits of this generalization are not fully explored. It remains unclear how robust these head locations are across different tasks, classes, or types of spurious correlation.

**Questions:**

See weakness

---

> ### Author Response · Authors · 2025-11-17
>
> Dear reviewer afvc,
>
> Thank you for your time in providing constructive feedbacks and comments. We are glad that you found our work to be clever and well motivated. We will address your concerns below.
>
> **1. Clarity**
>
> May we ask the reviewer to kindly point out the sections where it is difficult to understand, so that we can clarify any confusion or doubts on our work?
>
> **2. Generalizability of Located Heads**
>
> The reviewer is correct in pointing out that the “locate” step is dataset dependent. We also like to remind the reviewer that **only the spurious heads** are reused since the target concept (e.g. Waterbirds, CounterAnimal) varies across datasets. The reason why this is the case is due to the common concept space. In Genderbias-VL and FairFace, both of them share the same spurious concept: gender, and for Waterbirds and CounterAnimal: background. Sharing between different concept space (Waterbirds and Genderbias-VL) would not make any sense here.
>
> Once again, thank you for your time and we kindly ask if the reviewer could specify any remaining doubts or questions regarding our work. We will be more than happy to assist you. Otherwise, we hope we have addressed your concerns and we would appreciate if you could increase the scores accordingly.

---

> ### Author Response · Authors · 2025-11-27
> **Kind reminders**
>
> Dear Reviewer afvc,
>
> We sincerely thank the reviewer once again for their careful reading and constructive feedback, which have helped us significantly improve the quality and clarity of this work. We have addressed all raised concerns in detail in the rebuttal and corresponding revisions. We kindly invite the reviewer to reassess the manuscript in light of these changes, and if the responses have satisfactorily resolved the concerns, we would greatly appreciate reconsideration of the scores accordingly.

---

### Official Review · Reviewer_MWs1 · 2025-11-02

**Soundness:** 2
**Presentation:** 2
**Contribution:** 2
**Rating:** 2
**Confidence:** 4

**Summary:**

This paper introduces Locate-Then-Correct (LTC), a framework for debiasing CLIP by directly analyzing and modifying its attention heads. The key idea is to identify which attention heads are responsible for spurious correlations and correct them without retraining the model.

Specifically, LTC first identifies dominant attention heads that have a strong influence on model predictions by constructing a matrix V that measures each head’s direct effect across samples. Then, by contrasting correctly and incorrectly classified confounding samples (e.g., “waterbirds on land background”), the framework isolates which heads are spuriously correlated with confounding factors and which encode true task-relevant signals. Finally, LTC performs targeted correction: mean ablation, which replaces activations of spurious heads with their dataset mean to suppress bias, and knowledge injection, which reinforces task-relevant heads by projecting them onto text-derived feature directions from an LLM.

Experiments on Waterbirds, CounterAnimal, and GenderBias-VL show gains in bias reduction. Visual and textual analyses further confirm that LTC’s adjustments target semantically meaningful attention heads, offering an interpretable way to improve fairness in CLIP.

**Strengths:**

1. The method enhances interpretability by quantifying how much each attention head contributes to spurious predictions, revealing which internal components drive biased behavior. Moreover, by visualizing these heads, it allows clear inspection of which image regions lead to incorrect or biased decisions.
2. The proposed method introduces minimal modification to the original CLIP architecture, avoiding retraining and preserving the model’s original zero-shot capabilities.
3. The approach demonstrates consistent improvements across multiple datasets and bias types, showing strong generalizability and robustness of the framework.

**Weaknesses:**

1. My main concern is that the paper insufficiently discusses related work and does not provide a comparison in the table. A large body of recent literature has explored bias identification and mitigation in vision–language models, including both training-based and training-free approaches [1–9]. While the paper contrasts its approach with a few baselines, a more in-depth comparison is needed. For instance, B2T [1] mitigates bias by adding bias-related keywords to prompts, but such recent approaches are not clearly contrasted with LTC.
2. Presentation
    - The description of input supervision required for each method should be clarified. It is ambiguous whether baselines and LTC assume access to spurious attribute labels, target labels, or validation-only supervision.
    - Many important details are deferred to the appendix, which makes the paper harder to follow. For instance, Section 5.2 references about Table 6 and Figure 10 located in appendix.

[1] Kim et al., Discovering and Mitigating Visual Biases through Keyword Explanation, CVPR 2024.

[2] Gerych et al., BendVLM: Test-Time Debiasing of Vision-Language Embeddings, NeurIPS 2024.

[3] Dehdashtian et al., FairerCLIP: Debiasing CLIP's Zero-Shot Predictions using Functions in RKHSs, ICLR 2024.

[4] Jung et al., A Unified Debiasing Approach for Vision-Language Models across Modalities and Tasks, NeurIPS 2024.

[5] Phan et al., Controllable Prompt Tuning For Balancing Group Distributional Robustness, ICML 2024.

[6] Jang et al., Target Bias Is All You Need: Zero-Shot Debiasing of Vision-Language Models with Bias Corpus, ICCV 2025.

[7] Lu et al., Mitigating Spurious Correlations in Zero-Shot Multimodal Models, ICLR 2025.

[8] Yang et al., Debiasing Vison-Language Models with Text-Only Training, arXiv preprint.

[9] Zhu et al., Project-Probe-Aggregate: Efficient Fine-Tuning for Group Robustness, CVPR 2025.

**Questions:**

1. Line 352. What is the precise difference between LTC and LTC*? Is the distinction based on which dataset is used (e.g., training vs. validation set), or does it involve a different optimization procedure?
2. Line 355. The description of LTC-JTT is somewhat vague. Is the only modification that CLIP is replaced with a JTT-trained CLIP model, or are there additional differences in how LTC is applied during or after training?
3. Line 351 & 358. It seems that all methods use zero-shot–inferred group labels for the training set and ground-truth group labels for the validation set. Please confirm if this interpretation is correct and specify any exceptions or differing configurations across baselines.

---

> ### Author Response · Authors · 2025-11-17
> **Clarifications on reviewer's concerns (Part 1)**
>
> Dear reviewer MWs1,
>
> Thank you for taking the time in reading and providing constructive feedbacks! We will enumerate our responses to your comments below.
>
> **1. Lack of comparison with related debiasing works**
>
> Thank you for mentioning the works for comparison. However, we would like to point out that despite these works being relevant to debiasing of Vision-Language Models, all of them are dissimilar with our LTC framework. For example, for the works that are training-free, they can be categorised as either performing **debiasing on the global embedding (image or text) or implementing some form of prompt optimization on the text captions**, this is in contrast to our debiasing only on **localized** attention heads. For the training methods, examples include optimizing for group robustness or group sample balance, we do not introduce any changes to training methods, but rather adopt LTC as a post-hoc debiasing on top of a training method (in this case JTT) to show that it can be easily adaptable towards trained classifier. **More importantly, not all of these methods listed here can be used without access to group labels**.
>
> We list a quick summary of the comparisons, enumerated in the same order of the related works mentioned by the reviewer.
>
> - LTC vs B2T: LTC pinpoints specific ViT heads encoding spurious vs target signals and edits them (mean-ablation + head-local knowledge injection), yielding mechanistic interpretability and causal effects—**beyond keyword prompting**.
>
> - LTC vs BendVLM: LTC is component-level (per head), **not a global embedding transform**, and does not require a labeled reference set at test-time; it can run with zero-shot inferred group labels (and optionally a small val set when reported as LTC*).
>
> - LTC vs FairerCLIP: LTC requires no retraining of CLIP heads and performs localized projections/ablations only where contrastively identified, improving interpretability and avoiding **global distortions**.
>
> - LTC vs SFID: LTC provides head-level attribution and causal editing (mean-ablation of S-heads; injection on Y-heads) instead of **representation-level** substitution/imputation.
>
> - LTC vs CPT: LTC is inference-time and training-free on CLIP parameters; we additionally show it composes on top of training-based methods (JTT-LTC) to further improve worst-group accuracy.
>
>  - LTC vs Target Bias: LTC edits internal heads and demonstrates causal effects; not reliant on constructing or maintaining an **external bias corpus**.
>
> - LTC vs Lu et al.: LTC does not require estimating a **single global spurious direction**; it discovers which heads encode S vs Y by contrasting GN subsets and then edits only those heads.
>
> - LTC vs Text-only debiasing: Orthogonal—LTC works directly on the CLIP ViT with real images, needs **no synthetic text-only corpus**, and provides head-level explanations/edits.
>
> - LTC vs PPA: **PPA trains a classifier**; LTC is training-free on the CLIP backbone and yields mechanistic attribution by operating only on contrastively located heads.
>
> An obvious difference from these works is that none of these works utilize interpretability for debiasing, which is the main contribution of our work, combined with the knowledge to know which part of the model has been edited and what it represents. **We have included these works into our related works section in the revised manuscript and discuss the key differences from our work (highlighted in red).**
>
> Since we had evaluated works that perform global representation debiasing (Ortho-Cali and Roboshot). We choose to implement an additional evaluation against a prompt optimization method, B2T on Waterbirds and under the training-free setting.
>
> |ViT-B/16        |WG | Avg | Gap |
> |:--------------|------:|------:|------:|
> | B2T     | 60.2  | 72.1  | 11.9 |
> | LTC    | 73.3 | 74.6 | 1.3 |
>
> |ViT-L/14        |WG | Avg | Gap |
> |:--------------|------:|------:|------:|
> | B2T     | 69.8  | 79.2  | 9.4 |
> | LTC    | 75.5 | 84.0 | 8.5 |
>
> |ViT-H/14        |WG | Avg | Gap |
> |:--------------|------:|------:|------:|
> | B2T     | 65.7  | 72.7  | 5.0 |
> | LTC    | 77.4 | 80.4 | 3.0 |
>
> As seen above, LTC can consistently attain a higher average and WG score with a smaller gap as compared to B2T.

---

> ### Author Response · Authors · 2025-11-17
> **Clarifications on reviewer's concerns (Part 2)**
>
> **2. Presentation**
>
> Please refer to L352-353, where we mentioned not having access to spurious labels (this is needed to form the group labels), we only assume access to target label. We use a training set consisting of the images and target label to derive the spurious and target attention head postions, $P_Y$ and $P_S$. The spurious label in the training set is zero-shot predicted. We have clarified these details in the revised draft, highlighted in red.
>
> We also re-located the relevant figures and tables (Table 6 and Figure 10) to the main section for easier readability.

---

> ### Author Response · Authors · 2025-11-17
> **Clarifications on reviewer's questions**
>
> **3. Difference between LTC and LTC***
>
> The main difference is that LTC* to optimize for the best set of $P_Y$ and $P_S$ that would give the highest WG accuracy. In LTC, we assume that the contrastive solution, Eq. 10 can identify the correct attention heads that encodes the spurious and target attribute. The close performance between LTC and LTC* indicates that Eq.10 can indeed faithfully identify these heads.
>
> **4. JTT-LTC**
>
> Yes that is correct! LTC is directly applied as a post-hoc debiasing method ontop of the CLIP trained with JTT.
>
> **5 Group labels**
>
> We apologise for the confusion. In training-free baselines, we do not assume group labels, these are fully self-predicted. We only do so on training methods, following the assumption in Cont Adapter.
>
> Once again, we thank you for taking the time to review our work and providing helpful feedback! We hope that the above actions have addressed your concerns. If not, what further clarification or modifications could we make to improve your score?

---

> ### Author Response · Authors · 2025-11-27
> **Kind reminders**
>
> Dear Reviewer MWs1,
>
> We sincerely thank the reviewer once again for their careful reading and constructive feedback, which have helped us significantly improve the quality and clarity of this work. We have addressed all raised concerns in detail in the rebuttal and corresponding revisions. We kindly invite the reviewer to reassess the manuscript in light of these changes, and if the responses have satisfactorily resolved the concerns, we would greatly appreciate reconsideration of the scores accordingly.

---

### Meta-Review · Area_Chair_2FaC · 2026-01-06

**Summary:**

The primary and most consistent concern raised by multiple reviewers (MWs1, 2p5V) was the significant omission of relevant recent literature and a lack of comprehensive benchmarking against state-of-the-art debiasing methods. Specifically, reviewers noted the absence of comparisons with key training-free and test-time approaches (e.g., SFID, NeuronTune, EvA, and RefineViT), arguing that the proposed Locate-Then-Correct (LTC) framework must demonstrate superiority or distinct practical advantages over these existing global representation methods, not just offer mechanistic interpretability. Secondary concerns involved the clarity of the presentation, particularly regarding the theoretical justification for applying Causal Mediation Analysis to deep neural networks, the generalizability of the "located" heads across different tasks, and the limitation of the method to Transformer-based architectures.

**Reviewer Concerns:**

The authors successfully addressed queries regarding specific methodological clarifications, such as the dependency of the "locate" step on datasets, and the derivation of the inequalities in Equation 10. They also provided additional experimental comparisons against B2T, RefineViT, and SFID during the rebuttal phase, which partially mitigated the "missing literature" critique for some reviewers. However, the fundamental concern regarding the comparative scope remains outstanding; Reviewer 2p5V explicitly remained unconvinced by the authors' argument that mechanistic interpretability exempts the method from benchmarking against high-performing global representation methods. The theoretical validity of treating a non-linear neural network as a deterministic computation graph for causal analysis also remains a point of contention that was not fully resolved to the critical reviewers' satisfaction.

**Reviewer Scores:**

Reviewer dz5L explicitly stated they would maintain their score of 6 but would not champion the paper, indicating that further discussion would likely have solidified their stance as a passive support at best. Reviewer MWs1, who initially gave a strong reject due to missing literature, might have raised their score marginally to 4 given the authors' extensive inclusion of comparisons in the rebuttal, but likely not higher given the remaining presentation issues. Reviewer 2p5V would almost certainly have maintained their score of 4 or argued for a rejection, as they clearly stated in their post-rebuttal comment that the authors' justification for excluding broader baselines was unconvincing. Reviewer afvc, sitting at a 4, likely would have been swayed by 2p5V’s critique regarding the lack of distinct utility over SOTA, potentially keeping their score at 4 or downgrade to 2.

---

### Decision · Program_Chairs · 2026-01-26

Reject